# Symbal: Detecting Systematic Misalignments in Model-Generated Captions

## Abstract

Multimodal large language models (MLLMs) often introduce errors when generating image captions, resulting in misaligned image-text pairs. Our work focuses on a class of captioning errors that we refer to as systematic misalignments, where a recurring error in MLLM-generated captions is closely associated with the presence of a specific visual feature in the paired image. Given a vision-language dataset with MLLM-generated captions, our aim in this work is to detect such errors, a task we refer to as systematic misalignment detection. As our first key contribution, we introduce SYMBALBENCH, the first benchmark designed to evaluate automated methods for identifying systematic misalignments. SYMBALBENCH consists of 420 vision-language datasets from two domains (natural images and medical images) with annotated systematic misalignments. As our second key contribution, we present SYMBAL, which utilizes a structured, dual-stage setup with off-the-shelf foundation models to identify such errors and summarize results in natural language. SYMBAL exhibits strong performance on SYMBALBENCH, correctly identifying systematic misalignments in 63.8% of datasets, a nearly 4x improvement over the closest baseline. We supplement our evaluations on SYMBALBENCH with real-world evaluations, showing that SYMBAL can identify systematic misalignments in captions generated by an off-the-shelf MLLM. Ultimately, our novel task, benchmark, and method can aid users in auditing MLLM-generated captions and identifying critical failure modes, without requiring access to the underlying MLLM.

## 1 Introduction

Multimodal large language models (MLLMs) possess strong image captioning capabilities yet often introduce errors into generated captions (Sarto et al., 2025; Zhou et al., 2024; Liu et al., 2024). As a result, images and paired MLLM-generated captions may be *misaligned*, meaning that the generated text erroneously refers to features that are not visible in the image. For example, consider an MLLM that is tasked with generating a radiology report for an input medical image; in this setting, a misalignment may exist if the MLLM-generated report indicates the presence of cardiomegaly (a condition characterized by an enlarged heart) despite the image showing no evidence of this diagnosis. Misalignments can have severe consequences, particularly in safety-critical domains like medicine (Hardy et al., 2025; Nakaura et al., 2023).

Our work focuses on a critical yet previously-underexplored subclass of captioning errors that we refer to as *systematic misalignments*. We term a misalignment as *systematic* when a recurring error in MLLM-generated captions is closely associated with the presence of a specific visual feature in the paired image. For example, in the medical domain, incorrect diagnoses of cardiomegaly in the MLLM-generated reports may be strongly associated with the presence of pacemakers (an implanted medical device that regulates the heartbeat) in the corresponding image (Sourget et al., 2025; Kumar et al., 2025). Systematic misalignments are a particularly egregious class of errors because they often arise due to spurious correlations or biases learned by MLLMs during training. As a result, systematic misalignments typically involve features that frequently co-occur in the real-world yet are not deterministically linked; for instance, while cardiomegaly and pacemakers do co-occur frequently, the presence of a pacemaker in a medical image does not necessarily imply that the patient has

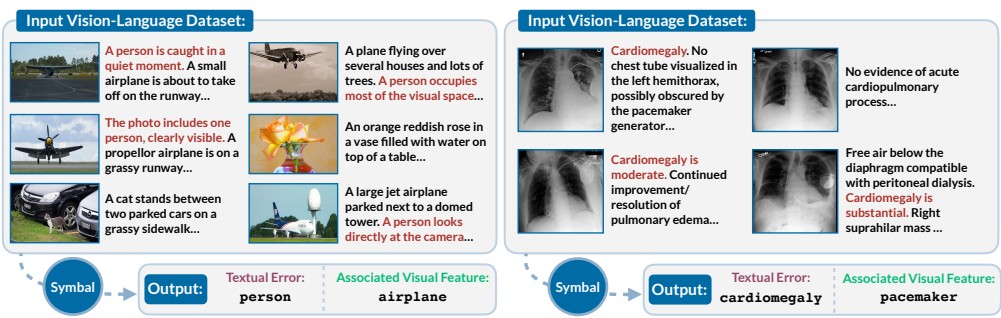

Figure 1: Given an input vision-language dataset with MLLM-generated captions, the *systematic misalignment detection task* involves identifying recurring textual errors and associated visual features. Here, we provide image-caption pairs from two datasets in SYMBALBENCH with expected outputs.

cardiomegaly. Thus, errors associated with systematic misalignments may seem highly plausible and are consequently challenging to detect.

In this work, we introduce the *systematic misalignment detection* task with the goal of leveraging automated approaches to identify this challenging class of captioning errors. A method that aims to solve the systematic misalignment detection task will accept as input a vision-language dataset, which consists of images paired with free-form MLLM-generated captions. Then, as output, the method must identify textual errors (e.g. "cardiomegaly" in the previous example) that are systematically associated with visual features (e.g. "pacemaker" in the previous example).

Addressing the systematic misalignment detection task with automated methods is challenging for the following two reasons. First, there are no existing benchmarks for comprehensively evaluating methods on their ability to discover systematic misalignments. Second, vision-language datasets provided as input to automated methods are often large in size with thousands of image-text pairs; identifying global error patterns from such datasets is nontrivial, especially since the size of such datasets exceeds the reasoning capabilities of even state-of-the-art models.

To address these challenges, we introduce the following contributions in this work:

- We introduce SYMBALBENCH, the first benchmark designed to evaluate systematic misalignment detection methods. SYMBALBENCH consists of 420 vision-language datasets from two domains (natural images and medical images) with known systematic misalignments. Each dataset is paired with a ground-truth annotation indicating the erroneous textual fact and associated visual feature; methods are then evaluated on their ability to accurately identify the annotated misalignment. SYMBALBENCH includes both reference-free and reference-based variants for each dataset as well as provides support for both open-ended and closed-ended prediction.

- We propose SYMBAL, an automated approach for detecting systematic misalignments in MLLM-generated captions.[1] Our key insight is to structure the systematic misalignment detection task into two stages, with each stage comprised of individual subtasks. The first stage of SYMBAL focuses solely on identifying recurring textual errors in captions; to this end, SYMBAL clusters textual facts based on semantic similarity, scores each cluster by degree of misalignment with paired images, and summarizes the top-ranked cluster into a single unifying concept. The second stage of SYMBAL then identifies the associated visual feature by clustering images paired with erroneous captions, scoring each image cluster by degree of misalignment with the identified textual error, and summarizing the top-ranked image cluster into a single unifying concept.

We evaluate SYMBAL using SYMBALBENCH, analyzing a range of possible approaches for addressing each subtask. Across the challenging reference-free, open-ended setting of SYMBALBENCH, the best configuration of SYMBAL correctly identifies the systematic misalignment in 63.8% of datasets. SYMBAL exhibits a nearly 4x improvement over the closest baseline, demonstrating the utility of our dual-stage, structured approach for addressing the systematic misalignment detection task.

---

[1]The acronym SYMBAL refers to **sy**stematic **m**isalignment detection **b**etween im**a**ges and **l**anguage.

Finally, we supplement our evaluations on SYMBALBENCH with real-world evaluations, surfacing previously-unknown systematic misalignments in captions generated by an off-the-shelf MLLM.

Ultimately, we hope that our novel task, benchmark, and method can (1) help users identify systematic captioning errors even *without access to the underlying MLLM*, a particularly important use-case as image datasets with MLLM-generated captions become widely available, and (2) assist model developers with understanding and mitigating failure modes in trained MLLMs.

## 2 RELATED WORK

We build on three research areas: (1) *sample-level misalignment detection* methods that identify captioning errors at the per-sample level; (2) *systematic error detection* methods that summarize global trends in prediction errors; and (3) methods for *describing large datasets in natural language*.

**Sample-Level Misalignment Detection:** Given an input data sample consisting of an image and a model-generated caption, one line of recent work has focused on developing metrics that measure image-caption alignment using numeric scores. Examples include reference-free metrics like CLIP-Score (Hessel et al., 2021) and PAC-S (Sarto et al., 2023), which do not require the existence of ground-truth captions; on the other hand, reference-based metrics such as BLEU (Papineni et al., 2002), ROUGE (Lin, 2004), CIDEr (Vedantam et al., 2015), METEOR (Banerjee & Lavie, 2005), and RefCLIPScore (Hessel et al., 2021) make use of ground-truth captions. The utility of such metrics is typically evaluated using image-caption benchmarks with human-annotated quality judgments (e.g. FLICKR8K-Expert (Hodosh et al., 2013), Pascal-50S (Vedantam et al., 2015), ReXVal (Yu et al., 2023)) or known model-injected errors (e.g. FOIL (Shekhar et al., 2017), ReXErr (Rao et al.)).

Several recent works have extended numeric scoring strategies by proposing interpretable metrics, which are capable of identifying the specific features in model-generated captions that are incorrect with respect to the image. Examples include reference-based metrics like CHAIR (Rohrbach et al., 2018), ALOHa (Petryk et al., 2024), and GREEN (Ostmeier et al., 2024) as well as reference-free metrics like FLEUR (Lee et al., 2024). Our work draws inspiration from these studies by also prioritizing interpretability; our method SYMBAL not only detects whether captioning errors are present but also provides users with a natural language output indicating the erroneous textual facts and associated visual cues. However, our study exhibits a key distinction from this line of work: whereas these metrics evaluate a single image and its paired model-generated caption, our work instead focuses on detecting *global*, systematic trends in captioning errors.

**Systematic Error Detection:** Due to visual biases or spurious correlations learned during training, machine learning models often make systematic prediction errors at test time. Selected examples in the classification setting noted by prior works include (1) an object recognition model that can correctly classify cows in pastoral settings yet demonstrates high error rates when cows are in beach settings (Beery et al., 2018) and (2) a pneumothorax detection model that achieves radiologist-level overall accuracy yet demonstrates high error rates when chest tubes, a medical device used for treatment, are absent (Oakden-Rayner et al., 2020).

A recent line of work has explored the development of automated methods for identifying systematic errors in classification settings. Given a validation dataset with images, model predictions, and ground-truth labels, these methods identify specific visual features (e.g. the beach background or the absence of tubes in the above examples) that are associated with higher error rates (Eyuboglu et al., 2022; Jain et al., 2023; Sohoni et al., 2020; Varma et al., 2024). Our work shares a similar goal in identifying systematic error patterns; however, we extend beyond the classification setting to the image captioning setting, where input datasets consist of images and paired model-generated captions. The inclusion of free-form text in input datasets presents an added level of complexity in comparison to labels; also, we explicitly consider settings where ground-truth captions are unavailable.

**Describing Datasets with Natural Language:** Several works have explored the challenge of describing patterns in data with natural language (Burgess et al., 2025); in particular, recent studies have generated natural language descriptions (i) summarizing differences given two input datasets (Dunlap et al., 2024; Zhong et al., 2022) and (ii) summarizing model prediction errors given classification datasets with labels (Eyuboglu et al., 2022; Menon & Srivastava, 2024; Kim et al., 2024). Our work also involves summarizing dataset-level patterns with natural language; however, we focus specifically on systematic misalignment detection, where datasets consist of images and paired captions.

## 3 Task Definition: Systematic Misalignment Detection

In this section, we formally introduce the systematic misalignment detection task. Consider a vision-language dataset $\mathcal{D} = \{(V_i, T_i)\}_{i=1}^{N}$ consisting of images $V$ paired with free-form, machine-generated text $T$. For example, dataset $\mathcal{D}$ may consist of chest X-rays $V$ paired with MLLM-generated radiology reports $T$. We will express each text sample $T_i$ as a collection of textual facts $T_i = \{f_1^i, f_2^i, ..., f_{n_i}^i\}$ and each image $V_i$ as a collection of visual features $V_i = \{g_1^i, g_2^i, ..., g_{m_i}^i\}$.

Dataset $\mathcal{D}$ may include misaligned samples, where text $T_i$ does not accurately describe the content of the paired image $V_i$. We consider a pair $(V_i, T_i)$ to be misaligned if there exists at least one erroneous textual fact $f_k^i \in T_i$ that does not accurately describe any visual feature $g_j^i \in V_i$. Misalignments are particularly egregious when they occur in a *systematic* fashion, meaning that an erroneous textual fact $f$ is repeatedly associated with the presence of a visual feature $g$ throughout a dataset. For instance, in the medical imaging example discussed earlier, perhaps incorrect diagnoses of cardiomegaly in MLLM-generated reports are strongly associated with the presence of a pacemaker in the corresponding chest X-rays; this suggests the existence of a systematic misalignment between reports containing the erroneous textual fact $f = cardiomegaly$ and images containing the visual feature $g = pacemaker$.

Thus, given a vision-language dataset $\mathcal{D}$, the goal of the **systematic misalignment detection** task is to discover textual errors $f$ that are systematically associated with visual cues $g$. A method $\mathcal{M} : \mathcal{D} \to (\hat{f}, \hat{g})$ that aims to solve the systematic misalignment detection task will accept dataset $\mathcal{D}$ as input; we note here that datasets may be large in size, consisting of thousands of image-text pairs. Then, method $\mathcal{M}$ will predict $(\hat{f}, \hat{g})$ as output, indicating the discovered textual error $\hat{f}$ and associated visual feature $\hat{g}$; here, both $\hat{f}$ and $\hat{g}$ will be expressed in text.

We consider two possible variants of input dataset $\mathcal{D}$: (1) a *reference-free* variant, where each sample in dataset $\mathcal{D} = \{(V_i, T_i)\}_{i=1}^{N}$ consists of an image $V_i$ paired with machine-generated text $T_i$, and (2) a *reference-based* variant, where each sample in dataset $\mathcal{D} = \{(V_i, T_i, C_i)\}_{i=1}^{N}$ consists of an image $V_i$, machine-generated text $T_i$, and a ground-truth reference caption $C_i$. We also consider two possible variants for the output of method $\mathcal{M}$: (1) *closed-ended*, where $\mathcal{M}$ must select from a list of possible options for the erroneous textual fact as well as a list of possible options for the associated visual feature, and (2) *open-ended*, where $\mathcal{M}$ must predict the misalignment without provided options. In combination, these variants comprise four possible experimental settings for the systematic misalignment detection task, of which the reference-free open-ended setting is most reflective of real-world use-cases.

## 4 Benchmark: SymbalBench

In this section, we introduce SymbalBench, the first benchmark designed to evaluate systematic misalignment detection methods. SymbalBench consists of a total of 420 vision-language datasets with known systematic misalignments. Each dataset $\mathcal{D}$ in SymbalBench is paired with a ground-truth annotation $(f, g)$ indicating the erroneous textual fact $f$ and associated visual feature $g$. Given $\mathcal{D}$ as input, method $\mathcal{M}$ is evaluated on its ability to accurately identify the annotated misalignment.

### 4.1 Benchmark Design

In order to create vision-language datasets with known systematic misalignments, we (1) obtain a high-quality base dataset with images and paired text, (2) predefine a systematic misalignment $(f, g)$, and (3) inject the erroneous textual fact $f$ into the base dataset such that a strong association exists with visual feature $g$. We then repeat this procedure across a wide range of possible options for $f$ and $g$. Importantly, our procedure is fully automated, enabling our benchmark-creation method to scale easily to diverse domains and modalities in future work. Below, we discuss these three steps in detail:

1. **Obtaining a base dataset.** We begin by obtaining an off-the-shelf vision-language dataset with high-quality samples. We consider two options for the base dataset: COCO (2017 val split) (Lin et al., 2015) and MIMIC-CXR (test split) (Johnson et al., 2019a). COCO consists of natural images depicting common objects from 80 categories. After preprocessing, the base dataset includes a total of 4349 images with associated captions. MIMIC-CXR consists of chest X-rays

and associated radiologist reports obtained from the Beth Israel Deaconess Medical Center. After preprocessing, the base dataset includes 2233 images, each paired with the "Impressions" section of the corresponding report. In the reference-based setting, we also include a ground-truth caption $C_i$ alongside each image-text pair $(V_i, T_i)$ in the base dataset.

2. **Predefining a systematic misalignment.** Given a base dataset, we predefine a systematic misalignment consisting of a textual fact $f$ and associated visual feature $g$. Predefined misalignments are meant to emulate those that are likely to emerge when using real-world, off-the-shelf MLLMs to generate captions. For COCO, we sample $f$ and $g$ from the set of 80 object categories present in the dataset. For MIMIC-CXR, we sample $f$ from a set of five disease categories (cardiomegaly, pneumothorax, atelectasis, pleural effusion, and edema) and $g$ from a set of five medical devices (pacemaker, chest tube, endotracheal tube, surgical clips, sternotomy wires). [2]

3. **Injecting the predefined systematic misalignment.** We insert the erroneous textual fact $f$ into text samples in the base vision-language dataset such that a strong association exists between text containing $f$ and images containing visual feature $g$. The strength of the association is controlled using Cramer's V scores. We then format each inserted fact $f$ as a natural language sentence.

We repeat this procedure across a range of possible options for $f$ and $g$, yielding 420 vision-language datasets with annotated systematic misalignments. Additional details are in Appendix A and B.

### 4.2 Benchmark Evaluation

In the closed-ended setting, a systematic misalignment detection method $\mathcal{M}$ is tasked with predicting $f$ and $g$ by selecting from a set of provided options. For datasets derived from COCO, we provide 80 options for both $f$ and $g$ representing object categories. For datasets derived from MIMIC-CXR, we provide 5 options for $f$ representing disease categories and 5 options for $g$ representing devices.

For each dataset $\mathcal{D}$ with ground-truth label $(f, g)$ and prediction $(\hat{f}, \hat{g})$, we count the prediction as accurate if the top-K predictions for $\hat{f}$ include $f$ and the top-K predictions for $\hat{g}$ include $g$. In open-ended settings, we leverage LLM-as-a-Judge with Llama3.3-70B to evaluate equality (Grattafiori et al., 2024). Overall performance on SYM-BALBENCH is measured with Accuracy@K, computed as the percentage of the 420 datasets in SYMBALBENCH where the prediction is accurate.

## 5 Our Approach: Symbal

The systematic misalignment detection task is made challenging by the fact that vision-language datasets may be complex and large in size; identifying global error patterns from such datasets is nontrivial. In this section, we address this challenge with our approach SYMBAL, which structures the systematic misalignment detection task into two stages. Each stage is comprised of three individual subtasks: grouping, scoring, and summarizing. Sections 5.1 and 5.2 discuss the two stages in detail.

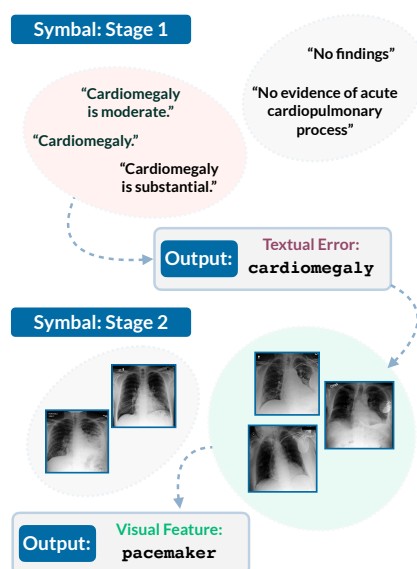

Figure 2: SYMBAL detects systematic misalignments with a 2-stage procedure.

### 5.1 Stage 1: Detecting Erroneous Textual Facts

The first stage of SYMBAL predicts the erroneous textual fact $\hat{f}$ by (1) grouping semantically-similar facts that occur consistently throughout the dataset, (2) scoring each group of facts by degree of

---

[2] We define these options for $f$ and $g$ due to the fact that medical imaging models often learn spurious associations between medical devices and disease categories, as documented in prior work (e.g. Oakden-Rayner et al. (2020)); thus, our predefined misalignments are highly plausible in real-world, model-generated reports.

misalignment with paired images, (3) and summarizing the top-ranked group of facts into a single unifying concept $\hat{f}$. The three subtasks associated with Stage 1 are detailed below:

- **Grouping semantically-similar facts:** As defined in Section 3, we first express each text sample $T_i$ as a collection of textual facts $T_i = \{f_1^i, f_2^i, ..., f_n^i\}$ by splitting captions at the sentence level. We then identify clusters of semantically-similar facts that occur in $\mathcal{D}$; for example, in the medical imaging example discussed earlier, perhaps one such cluster will contain sentences from radiology reports that discuss the presence of cardiomegaly. To this end, we aggregate all textual facts in $\mathcal{D}$, forming the set $\bigcup_{i=1}^N T_i = \{f_k^i : i = 1, ..., N; k = 1, ..., n_i\}$. Each textual fact in this set is encoded using a text embedding model; then, embeddings are clustered using spherical K-Means, where the number of clusters is selected automatically using Silhouette distance.

- **Scoring groups by degree of misalignment:** Next, we score each cluster by computing the mean degree of alignment between constituent textual facts and paired images. Based on methods proposed in prior work (Hessel et al., 2021; Dunlap et al., 2024; Chen et al., 2024a), we consider three possible scoring mechanisms for measuring alignment between a given textual fact and its paired image: (1) *embedding scorer*, which computes embeddings for the text and image modalities and measures alignment as the cosine similarity, (2) *text-only scorer*, which generates a caption for the image and tasks an LLM with determining if the textual fact is accurate with respect to the caption, and (3) *vision-language scorer*, where a MLLM is provided both the image and the textual fact as input and tasked with determining if the textual fact is accurate. Low scores suggest that a large proportion of textual facts in the cluster are misaligned with respect to their paired images.

- **Summarizing the top-ranked group:** Given the alignment scores computed in the previous step, we identify the cluster exhibiting the highest degree of misalignment, which we will refer to as $C_{text}$. Then, we consider two summarization mechanisms for identifying the unifying concept shared by textual facts in $C_{text}$: (1) *embedding summarizer*, which selects the closed-ended option with the highest embedding-based cosine similarity to textual facts in $C_{text}$, and (2) *text-only summarizer*, where an LLM is provided a list of textual facts in $C_{text}$ and tasked with identifying the unifying concept. The embedding summarizer is only utilized in closed-ended settings.

The final output of the summarizer is the predicted erroneous textual fact $\hat{f}$; for example, in the medical example discussed earlier, the predicted textual fact may be $\hat{f} = cardiomegaly$. In Section 6.1, we evaluate the role of various text embedding models, alignment scorers, and summarizers.

## 5.2 Stage 2: Detecting Associated Visual Features

We now proceed to the second stage of SYMBAL, which predicts the visual feature $\hat{g}$ by (1) grouping semantically-similar images paired with text containing fact $\hat{f}$, (2) scoring each group of images by degree of misalignment with paired text, and (3) summarizing the top-ranked group of images into a single unifying concept $\hat{g}$. The three subtasks associated with Stage 2 are detailed below:

- **Grouping semantically-similar images:** We begin by identifying all images $V_i \in \mathcal{D}$ containing at least one paired textual fact in cluster $C_{text}$ (i.e. where $f_k^i \in C_{text}$ for some $k$). Each image in this set is encoded using an image embedding model; then, embeddings are clustered using spherical K-Means, where the number of clusters is selected automatically using Silhouette distance.

- **Scoring groups by degree of misalignment:** Next, we score each cluster by computing the mean degree of misalignment between images and paired textual facts in $C_{text}$. We consider the same scoring mechanisms as in Stage 1. Low scores suggest that a large proportion of images in the cluster are misaligned with fact $\hat{f}$.

- **Summarizing the top-ranked group:** Given the alignment scores computed in the previous step, we identify the cluster exhibiting the highest degree of misalignment, which we will refer to as $C_{image}$. Then, we consider three summarization mechanisms for identifying the unifying concept shared by images in $C_{image}$: (1) *embedding summarizer*, which selects the closed-ended option with the highest embedding-based cosine similarity to images in $C_{image}$, (2) *text-only summarizer*, where a caption is generated for each image in $C_{image}$ and an LLM is tasked with identifying the unifying concept, and (3) *vision-language summarizer*, where an MLLM is provided with images in $C_{image}$ and tasked with identifying the unifying concept.

Table 1: We consider the role of various text embedding models, alignment scorers, and summarizers on the performance of Stage 1 of SYMBAL. Here, VL refers to the vision-language scorer and MG-27B refers to MedGemma-27B.

| | Text Embedding | Alignment Scorer | Summarizer | Reference-Free | | | | Reference-Based | | | |
| | | | | Closed-Ended | | Open-Ended | | Closed-Ended | | Open-Ended | |
| | | | | Acc@1 | Acc@5 | Acc@1 | Acc@5 | Acc@1 | Acc@5 | Acc@1 | Acc@5 |
|---|---|---|---|---|---|---|---|---|---|---|---|
| Natural | Qwen3-8B | VL (Qwen-72B) | Text (Qwen-72B) | **93.9** | **94.4** | **92.8** | **94.2** | 84.4 | 85.3 | 80.8 | 82.8 |
| | OpenCLIP | VL (Qwen-72B) | Text (Qwen-72B) | **93.9** | **94.4** | **92.8** | 93.9 | **87.2** | **88.6** | **86.1** | **87.8** |
| | Qwen3-8B | Text (Qwen-72B) | Text (Qwen-72B) | 83.9 | 85.8 | 82.8 | 85.0 | 84.2 | 85.3 | 81.9 | 83.9 |
| | OpenCLIP | Text (Qwen-72B) | Text (Qwen-72B) | 66.1 | 68.6 | 64.2 | 67.2 | 70.6 | 72.2 | 67.5 | 71.4 |
| Medical | XRayCLIP | Text (MG-27B) | Text (MG-27B) | **58.3** | – | **51.7** | 75.0 | **100.0** | – | 88.3 | 95.0 |
| | XRayCLIP | Text (MG-27B) | Text (Qwen-72B) | 56.7 | – | **51.7** | 73.3 | **100.0** | – | **100.0** | **100.0** |
| | XRayCLIP | Text (Qwen-72B) | Text (MG-27B) | 31.7 | – | 26.7 | 58.3 | 98.3 | – | 90.0 | 93.3 |
| | MedSigLIP | Text (MG-27B) | Text (MG-27B) | 45.0 | – | 30.0 | 53.3 | **100.0** | – | 83.3 | **100.0** |

Table 2: We consider the role of various image embedding models, alignment scorers, and summarizers on the performance of Stage 2 of SYMBAL. Here, VL refers to the vision-language scorer, Emb. refers to the embedding scorer, and MG-27B refers to MedGemma-27B.

| | Img Embedding | Alignment Scorer | Summarizer | Reference-Free | | | | Reference-Based | | | |
| | | | | Closed-Ended | | Open-Ended | | Closed-Ended | | Open-Ended | |
| | | | | Acc@1 | Acc@5 | Acc@1 | Acc@5 | Acc@1 | Acc@5 | Acc@1 | Acc@5 |
|---|---|---|---|---|---|---|---|---|---|---|---|
| Natural | OpenCLIP | VL (Qwen-72B) | Text (Qwen-72B) | 52.2 | 71.1 | **49.7** | **69.7** | 42.5 | **60.3** | 41.9 | 52.2 |
| | OpenCLIP | Emb. (OpenCLIP) | VL (Qwen-72B) | **53.9** | **71.4** | 48.1 | 63.9 | **45.6** | 57.8 | 42.5 | 55.6 |
| | OpenCLIP | Emb. (OpenCLIP) | Text (Qwen-72B) | 48.6 | 67.5 | 47.8 | 62.8 | 45.3 | 59.2 | **43.9** | **55.8** |
| | OpenCLIP | VL (Qwen-72B) | VL (Qwen-72B) | **53.9** | 70.6 | 45.8 | 62.5 | 44.4 | 59.2 | 38.9 | 52.2 |
| Medical | XRayCLIP | Emb. (MedSigLIP) | VL (MG-27B) | **26.7** | – | 11.7 | **36.7** | 41.7 | – | 28.3 | 53.3 |
| | MedSigLIP | Emb. (MedSigLIP) | VL (MG-27B) | 23.3 | – | 11.7 | 31.7 | 40.0 | – | 25.0 | 46.7 |
| | OpenCLIP | Emb. (MedSigLIP) | VL (MG-27B) | 23.3 | – | **13.3** | 28.3 | 35.0 | – | 20.0 | 46.7 |
| | MedSigLIP | Emb. (XRayCLIP) | VL (MG-27B) | 25.0 | – | 10.0 | 28.3 | **50.0** | – | **33.3** | **60.0** |

The final output of the summarizer is the predicted visual feature $\hat{g}$; for example, in the medical example discussed earlier, the predicted visual feature may be $\hat{g} = pacemaker$. In Section 6.2, we evaluate the role of various image embedding models, alignment scorers, and summarizers.

## 6 RESULTS

We now evaluate SYMBAL on the systematic misalignment detection task. In Sections 6.1 and 6.2, we use SYMBALBENCH to analyze the choice of embedding models, alignment scorers, and summarizers. In Section 6.3, we perform end-to-end evaluations of the best configuration of SYMBAL, comparing with baselines and performing fine-grained analyses. Finally, in Section 6.4, we extend beyond SYMBALBENCH to real-world settings.

### 6.1 SYMBAL DETECTS ERRONEOUS TEXTUAL FACTS

We first evaluate the role of various text embedding models, alignment scorers, and summarizers on the performance of Stage 1 of SYMBAL, which aims to identify the erroneous textual fact given an input dataset $\mathcal{D}$ in SYMBALBENCH. The accuracy of predicted textual facts $\hat{f}$ is evaluated using Accuracy@1 and Accuracy@5.[3] Results are summarized in Table 1.

For the natural image datasets in SYMBALBENCH, Table 1 Upper demonstrates the performance of the top-four compositions, ranked by Accuracy@5 scores on the reference-free, open-ended setting. Our results show that the best-performing variant of SYMBAL (shown in Row 1 of Table 1 Upper) achieves strong performance, correctly identifying the erroneous textual fact in over 90% of SYMBALBENCH datasets in the reference-free configuration (Closed-Ended Acc@5 = 94.4, Open-Ended Acc@5 = 94.2) and over 80% of SYMBALBENCH datasets in the reference-based configuration (Closed-Ended Acc@5 = 85.3, Open-Ended Acc@5 = 82.8). Interestingly, we find that performance in reference-free settings is often substantially higher than performance in the reference-based setting, which is likely a result of the sparse information content often present in COCO reference captions. When considering the composition of SYMBAL, we note that the choice of the alignment scorer appears to be most important; the vision-language scorer substantially outperforms the text-only scorer with the same

---

[3]We do not report Accuracy@5 on closed-ended settings for medical datasets derived from MIMIC-CXR due to the fact that there are only five options provided. Thus, Accuracy@5 is trivially 1.0.

underlying model (Qwen2.5-72B). Given these results, we select the Qwen3-Embedding-8B text embedding model (Zhang et al., 2025), the vision-language alignment scorer with Qwen2.5-72B (Qwen et al., 2025), and the text-only summarizer with Qwen2.5-72B (Qwen et al., 2025) for all future SYMBAL evaluations on natural images.

For the medical image datasets in SYMBALBENCH, Table 1 Lower demonstrates the performance of the top-four compositions, ranked by Accuracy@5 scores on the reference-free, open-ended setting. Our results show that the best-performing variant of SYMBAL (shown in Row 1 of Table 1 Lower) correctly identifies the erroneous textual feature in over 50% of datasets in the reference-free configuration (Closed-Ended Acc@1 = 58.3, Open-Ended Acc@5 = 75.0) and over 95% of datasets in the reference-based configuration (Closed-Ended Acc@1 = 100.0, Open-Ended Acc@5 = 95.0). In contrast to the natural image datasets, we find that the reference-free configuration is substantially harder than the reference-based configuration, likely due to the complexity of medical image data; alignment scoring in this domain is challenging without access to reference text. We also note that a key advantage of SYMBAL is its ability to extend to specialized domains simply by interchanging constituent models with domain-specific versions; indeed, we find that the best-performing variant of SYMBAL leverages models that were trained on domain-specific radiology data. Given these results, we select the XRayCLIP-ViT-L text embedding model (Chen et al., 2024b), the text-only alignment scorer with MedGemma-27B (Sellergren et al., 2025), and the text-only summarizer with MedGemma-27B (Sellergren et al., 2025) for all future SYMBAL evaluations on medical images.

## 6.2 SYMBAL DETECTS ASSOCIATED VISUAL FEATURES

We next evaluate the role of various image embedding models, alignment scorers, and summarizers on the performance of Stage 2 of SYMBAL. We hold the composition of Stage 1 constant using results from Section 6.1. The accuracy of predicted visual features $\hat{g}$ is evaluated using Accuracy@1 and Accuracy@5. Results are summarized in Table 2.

For the natural image datasets in SYMBALBENCH, Table 2 Upper demonstrates the performance of the top-four compositions, ranked by Accuracy@5 scores on the reference-free, open-ended setting. Our results show that the best-performing variant of SYMBAL (shown in Row 1 of Table 2 Upper) correctly identifies the visual feature in approximately 70% of datasets in the reference-free configuration (Closed-Ended Acc@5 = 71.1, Open-Ended Acc@5 = 69.7) and over 50% of datasets in the reference-based configuration (Closed-Ended Acc@5 = 60.3, Open-Ended Acc@5 = 52.2). We observe that performance values in Table 2 are lower than 1, suggesting that identifying visual features that systematically occur with textual errors is substantially more challenging than identifying the textual error itself. We also observe that the best-performing variant of SYMBAL utilizes the same alignment scorer and summarizer as in Stage 1. Given these results, we select the OpenCLIP-ViT-H image embedding model (Ilharco et al., 2021), vision-language alignment scorer with Qwen2.5-72B (Qwen et al., 2025), and text-only summarizer with Qwen2.5-72B (Qwen et al., 2025) for all future SYMBAL evaluations on natural images.

For the medical image datasets in SYMBALBENCH, Table 2 Lower demonstrates the performance of the top-four compositions, ranked by Accuracy@5 scores on the reference-free, open-ended setting. Our results show that the best-performing variant of SYMBAL (shown in Row 1 of Table 2 Lower) correctly identifies the visual feature in over 25% of datasets in the reference-free configuration (Closed-Ended Acc@1 = 26.7, Open-Ended Acc@5 = 36.7) and over 40% of datasets in the reference-based configuration (Closed-Ended Acc@1 = 41.7, Open-Ended Acc@5 = 53.3). Our results suggest that identifying visual features in the medical domain is a particularly challenging task in both reference-free and reference-based settings, and consequently, the optimal composition of alignment scorers and summarizers differs markedly from those identified in Stage 1. Given these results, we select the XRayCLIP-ViT-L image embedding model (Chen et al., 2024b), embedding alignment scorer with MedSigLIP (Sellergren et al., 2025), and vision-language summarizer with MedGemma-27B (Sellergren et al., 2025) for all future SYMBAL evaluations on medical images.

## 6.3 SYMBAL DEMONSTRATES STRONG END-TO-END PERFORMANCE

Given an optimal composition of SYMBAL, we now perform end-to-end analyses across SYM-BALBENCH. Since our study proposes a novel task, there are no existing baselines for comparison. As a result, we compare the structured, dual-stage approach of SYMBAL to a single-

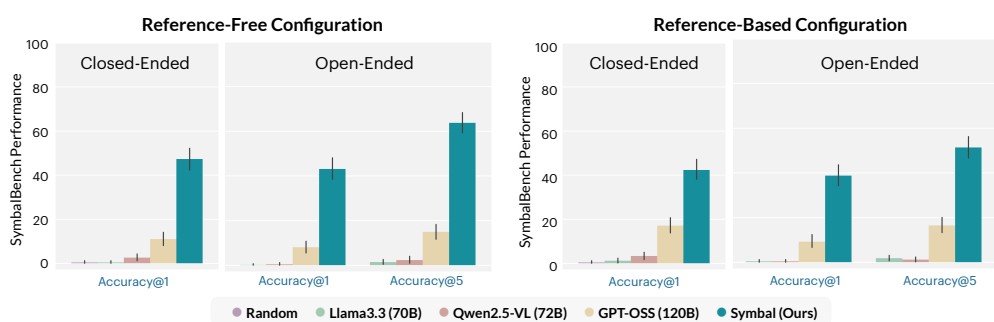

Figure 3: SYMBAL demonstrates strong end-to-end performance on SYMBALBENCH, substantially outperforming comparable baselines.

stage, direct-prompting method where each dataset $\mathcal{D}$ is directly provided to an off-the-shelf LLM in the form of a text prompt; the LLM is then instructed to output the erroneous textual fact and the associated visual feature. Three state-of-the-art LLMs are considered (i.e. Llama3.3 70B, Qwen2.5-VL 72B, and GPT-OSS 120B), selected to ensure a fair comparison with SYMBAL due to comparable parameter counts. As the token length of the direct prompts far surpasses the context window of these LLMs, we use only a sample of each dataset, ensuring that the final inference procedure requires no more compute resources than SYMBAL. In closed-ended settings, we also evaluate a random baseline, where $\hat{f}$ and $\hat{g}$ are randomly-selected options.

In Figure 3, we measure the extent to which SYMBAL can accurately predict *both* the textual fact $\hat{f}$ and the visual feature $\hat{g}$ across the four possible experimental settings associated with SYMBALBENCH. Results show that the systematic misalignment detection task is highly challenging in all four experimental settings, with several baselines generating few correct predictions. SYMBAL successfully identifies the systematic misalignment in up to 63.8% of datasets in SYMBALBENCH, with the highest performance observed in the reference-free, open-ended setting (Accuracy@5). SYMBAL outperforms the closest baseline (GPT-OSS 120B) across all experimental settings, with GPT-OSS 120B correctly identifying the misalignment in only 17.1% of SYMBALBENCH datasets in the best case. These results demonstrate that the structured, dual-stage approach utilized by SYMBAL provides substantial performance benefits over single-stage, direct prompting baselines. In Figure 4, we provide a stratified breakdown of SYMBAL performance. SYMBAL continues to outperform baselines across challenging subsets of SYMBALBENCH that exhibit (1) weak association between the textual error and visual feature as measured by Cramer's V scores and (2) small features.

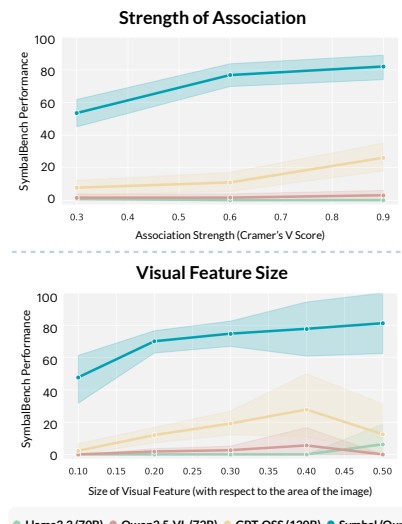

Figure 4: We report performance on SYMBALBENCH (reference-free, open-ended) stratified across association strengths and visual feature sizes. This focuses on natural image datasets.

## 6.4 SYMBAL EXTENDS TO REAL-WORLD SETTINGS

In this section, we further demonstrate the utility of SYMBAL by supplementing our evaluations on SYMBALBENCH with additional quantitative and qualitative analyses in real-world settings. Our results show that (1) SYMBAL can accurately surface systematic misalignments in captions generated by off-the-shelf MLLMs and (2) SYMBAL is a powerful tool for auditing vision-language datasets.

**SYMBAL can accurately surface systematic misalignments in captions generated by off-the-shelf MLLMs.** First, we use SYMBAL to analyze captions generated by four real-world off-the-shelf MLLMs: Llava1.5-7B (Liu et al., 2023), Llava1.5-13B (Liu et al., 2023), AyaVision-8B (Dash et al., 2025), and LlavaOneVision-7B (Li et al., 2024). We utilize each model to generate captions for

the COCO dataset (2017 val split); we then apply SYMBAL (reference-free, open-ended) to predict systematic misalignments $(\hat{f}, \hat{g})$.

We note that since ground-truth systematic misalignments in real-world settings are unknown, assessing the accuracy of results poses a challenge. Here, in order to address this issue, we validate identified systematic misalignments in two ways. First, we *qualitatively* validate the existence of SYMBAL-identified systematic misalignments with visual analysis. Second, we *quantitatively* validate whether a link between erroneous fact $\hat{f}$ and visual feature $\hat{g}$ truly exists; to this end, we measure whether model-generated captions are indeed more likely to include erroneous references to $\hat{f}$ when $\hat{g}$ is present compared to when $\hat{g}$ is absent. In order to perform this evaluation, we use a state-of-the-art open-set object detector (Minderer et al., 2024) to annotate the presence of $\hat{g}$ in each image, and we use our top-performing alignment scorer (vision-language scorer with Qwen-72B) to annotate erroneous references to $\hat{f}$ in each caption. In Appendix E, we demonstrate that automated annotations align closely with human judgments.

Below, we list several SYMBAL-identified systematic misalignments. Additional examples and evaluations are provided in Appendix E.

- In captions generated by Llava1.5-7B, SYMBAL detects that erroneous references to a handbag or a handbag on the ground ($\hat{f}$) in captions are often systematically associated with the presence of a bus ($\hat{g}$) in a scene, as shown in Figure 8 [Row 2]. Quantitatively, our analysis finds that erroneous references to a handbag in model-generated captions are indeed 3.1 times more likely when a bus is present in the image compared to when a bus is absent, validating the SYMBAL prediction.

- In captions generated by LlavaOneVision-7B, SYMBAL detects that erroneous references to text ($\hat{f}$) in captions are often systematically associated with the presence of a sign ($\hat{g}$) in a scene, as shown in Figure 9 [Row 2]. This finding suggests that LlavaOneVision-7B struggles with OCR capabilities, where the presence of text-based signage in an image is likely to result in errors in the generated caption. Quantitatively, our analysis finds that erroneous references to text in model-generated captions are indeed 4.6 times more likely when a sign is present in the image compared to when a sign is absent, validating the SYMBAL prediction.

**SYMBAL is a powerful tool for auditing open-source vision-language datasets.** Second, we use SYMBAL to analyze ShareGPT4V, an open-source image dataset with MLLM-generated captions commonly used as a pretraining dataset for vision-language models (Chen et al., 2023). We sample a subset of 10k image-caption pairs from the ShareGPT4V dataset, and we then apply SYMBAL (reference-free, open-ended) to predict systematic misalignments $(\hat{f}, \hat{g})$. SYMBAL identifies several systematic misalignments. For example, SYMBAL detects that erroneous references to a white tablecloth ($\hat{f}$) in captions are often systematically associated with the presence of a table, cake, and/or people ($\hat{g}$) in the scene, as shown in Figure 10 [Row 1]. Quantitatively, our analysis finds that erroneous references to a white tablecloth in model-generated captions are indeed 17.2 times more likely when a table is present in the image compared to when a table is absent, validating the SYMBAL prediction. Additional examples are provided in Appendix E.

As large-scale datasets like ShareGPT4V become increasingly prevalent, it becomes critical for users to be aware of potential systematic misalignments, as these errors can propagate to trained models. Specifically, if a dataset contains a systematic misalignment between erroneous textual fact $\hat{f}$ and visual feature $\hat{g}$, models trained on the dataset are likely to learn spurious correlations between $\hat{f}$ and $\hat{g}$, leading to prediction errors at test-time (Varma et al., 2024). Ultimately, knowledge of systematic misalignments can aid users with understanding limitations of datasets with MLLM-generated captions as well as aid model developers with improving performance of MLLMs.

## 7 DISCUSSION

In this work, we introduce the systematic misalignment detection task, which aims to identify textual errors in MLLM-generated captions that are systematically associated with visual features. We hope that our novel task, benchmark SYMBALBENCH, and method SYMBAL can help users audit MLLM-generated captions and identify failure modes, even without access to the underlying MLLM.

## REPRODUCIBILITY STATEMENT

Dataset preprocessing and implementation details are discussed in Appendix Sections A to E. We will make data associated with SYMBALBENCH and code associated with SYMBAL publicly-available at the conclusion of the anonymity period.

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

APPENDIX

CONTENTS

## A  IMPLEMENTATION DETAILS FOR SYMBALBENCH

SYMBALBENCH includes a total of 420 vision-language datasets, with 360 natural image datasets derived from COCO and 60 medical image datasets derived from MIMIC-CXR. Below, we provide extended implementation details for the natural image datasets:

1. **Obtaining a base dataset.** The base vision-language datasets in the natural image domain are derived from COCO (2017 val split), which consists of photographs depicting common objects (e.g. animals, food, furniture, etc.) in natural settings. Images are paired with object-level annotations as well as five human-written captions, with each caption typically consisting of a single sentence or phrase describing salient features in the image. In order to ensure that objects are clearly visible in the image, we exclude annotations for all small objects, defined as objects that take up less than 5% of the area of the image. After filtering out images with no remaining object-level annotations, we are left with a base dataset consisting of 4349 images and associated captions. We then compose a new two-sentence caption for each image by randomly sampling two captions from the provided list of five captions.

2. **Predefining a systematic misalignment.** We then predefine a set of systematic misalignments, each consisting of a textual fact $f$ and the associated visual feature $g$. We sample $g$ from the set of 80 object categories present in the dataset. Then, we sample $f$ from the set of 80 object categories (such that $f \neq g$) utilizing three possible sampling strategies: (1) *random*, where $f$ is sampled randomly, (2) *popular*, where $f$ is sampled from the list of the top-ten most popular objects in the COCO training set, and (3) *adversarial*, where $f$ is the object that most commonly co-occurs with $g$ in the COCO training set. These sampling strategies are motivated by prior work (Li et al., 2023) and are meant to capture a range of possible error patterns that may emerge in real-world MLLM-generated captions.

3. **Injecting the predefined systematic misalignment.** We insert the erroneous textual fact $f$ into captions in the base dataset, ensuring that a association exists between text containing $f$ and images containing visual feature $g$; this procedure ensures that the misalignment is *systematic*. Importantly, we ensure that feature $f$ is not already in the image-caption pair prior to injection. We consider three possible levels of association, as measured by Cramer's V: low association (Cramer's V = 0.3), moderate association (Cramer's V = 0.6), and high association (Cramer's V = 0.9). In order to format textual fact $f$ into a sentence, we generate 50 templates using GPT-4o, select a template at random, and insert $f$. We repeat this injection procedure for all possible choices of $f$ and $g$ in order to obtain 360 vision-language datasets $\mathcal{D}$ with known systematic misalignments.

Below, we provide extended implementation details for the medical image datasets:

1. **Obtaining a base dataset.** The base vision-language datasets in the medical image domain are derived from MIMIC-CXR (test split), which consists of chest X-rays and associated radiologist reports collected at Beth Israel Deaconess Medical Center. We preprocess the dataset by (1)

removing all images with non-frontal imaging views, (2) removing all images with missing "Impressions" sections in the paired report, and (3) removing all sentences in reports without "present" disease or anatomy entities, as identified by an off-the-shelf medical entity annotation tool (Delbrouck et al., 2024). After preprocessing, we are left with a base dataset consisting of 2233 images, each paired with the "Impressions" section of the corresponding report.

2. **Predefining a systematic misalignment.** We sample $f$ from a set of five disease categories selected from the commonly-used CheXpert annotation list (Irvin et al., 2019): cardiomegaly, pneumothorax, atelectasis, pleural effusion, and edema. We sample $g$ from a set of five medical devices: pacemaker, chest tube, endotracheal tube, surgical clips, sternotomy wires. We select these options for $f$ and $g$ since medical devices often co-occur with diseases, yet there is no deterministic, universal link. Models often learn spurious associations between devices and diseases as documented in prior work (Oakden-Rayner et al., 2020), meaning that such errors are highly plausible in MLLM-generated reports.

3. **Injecting the predefined systematic misalignment.** We insert the erroneous textual fact $f$ into reports in the base dataset, using Cramer's V to control the level of association with visual feature $g$. We use a combination of physician annotations, automated annotations from the CheXpert labeler (Irvin et al., 2019), and automated annotations from RadGraph-XL (Delbrouck et al., 2024) in order to identify whether or not $f$ and $g$ are present in the image-report pair prior to injection. In order to format textual fact $f$ into a sentence, we identify the 50 most frequently occurring sentences in the MIMIC-CXR training set that discuss the presence of $f$ and select a sentence from this list at random. We repeat this injection procedure for all possible choices of $f$ and $g$ in order to obtain 60 vision-language datasets $\mathcal{D}$ with known systematic misalignments.

In reference-based settings, we also include a ground-truth caption $C_i$ along with each image-text pair $(V_i, T_i) \in \mathcal{D}$. For natural image datasets derived from COCO, $C_i$ takes the form of a three-sentence caption combining the three human-written captions not originally selected as part of $T_i$. For medical image datasets derived from MIMIC-CXR, $C_i$ takes the form of the "Findings" and "Impressions" sections of the original physician-written radiology report. We emphasize that $T_i$ may contain errors as a result of the error-injection procedure detailed above; however, $C_i$ is always accurate.

In closed-ended settings, we provide a set of options for $f$ and $g$. For natural image datasets derived from COCO, we provide the following 80 options for $f$ and $g$: airplane, apple, backpack, banana, baseball bat, baseball glove, bear, bed, bench, bicycle, bird, boat, book, bottle, bowl, broccoli, bus, cake, car, carrot, cat, cell phone, chair, clock, couch, cow, cup, dining table, dog, donut, elephant, fire hydrant, fork, frisbee, giraffe, hair drier, handbag, horse, hot dog, keyboard, kite, knife, laptop, microwave, motorcycle, mouse, orange, oven, parking meter, person, pizza, potted plant, refrigerator, remote, sandwich, scissors, sheep, sink, skateboard, skis, snowboard, spoon, sports ball, stop sign, suitcase, surfboard, teddy bear, tennis racket, tie, toaster, toilet, toothbrush, traffic light, train, truck, tv, umbrella, vase, wine glass, zebra. For medical image datasets derived from MIMIC-CXR, we provide the following 5 options for $f$: cardiomegaly, pleural effusion, pneumothorax, edema, atelectasis. For medical image datasets derived from MIMIC-CXR, we provide the following 5 options for $g$: pacemaker, chest tube, endotracheal tube, surgical clips, sternotomy wires.

In open-ended settings, we determine if predictions are equivalent to the ground-truth by leveraging LLM-as-a-Judge. We use Llama3.3-70B in all experiments as the LLM, leveraging the `ollama` implementation with default parameters. The input prompt is provided below:

---

**LLM-as-a-Judge Evaluation Prompt**

You are given two short text phrases.
Model response: <predicted textual error or predicted visual feature>
Ground truth: <ground-truth textual error or ground-truth visual feature>

Your task is to determine if both phrases refer to the same visual feature. Please output 1 if both the model response and the correct answer refer to the same feature or 0 if the model response and the correct answer do not refer to the same feature. Do not provide anything other than the number in your response.

---

## B  SYMBALBENCH DESCRIPTIVE STATISTICS

In this section, we provide descriptive statistics summarizing the composition of SYMBALBENCH. SYMBALBENCH includes 420 vision-language datasets covering two domains (with 360 natural image datasets and 60 medical image datasets). In Table 3, we provide a list of all ground-truth systematic misalignments $(f, g)$ included in SYMBALBENCH.

Table 3: Here, we provide a list of all ground-truth systematic misalignments $(f, g)$ included in SYMBALBENCH.

| Erroneous Textual Fact $f$ | Visual Feature $g$ | Erroneous Textual Fact $f$ | Visual Feature $g$ | Erroneous Textual Fact $f$ | Visual Feature $g$ |
|---|---|---|---|---|---|
| surfboard | airplane | person | airplane | bottle | airplane |
| person | banana | chair | banana | car | banana |
| kite | bed | person | bed | chair | bed |
| person | bench | handbag | bench | oven | bench |
| hot dog | bicycle | person | bicycle | truck | bicycle |
| person | bird | wine glass | bird | book | bird |
| truck | boat | person | boat | bicycle | boat |
| toilet | book | cup | book | person | book |
| pizza | bottle | person | bottle | elephant | bowl |
| car | bowl | dining table | bowl | cat | broccoli |
| dining table | broccoli | car | broccoli | handbag | bus |
| frisbee | bus | person | bus | bicycle | cake |
| dining table | cake | chair | cake | fork | car |
| person | car | car | cat | umbrella | cat |
| person | cat | airplane | chair | person | chair |
| car | chair | bottle | couch | baseball glove | couch |
| person | couch | person | cow | cake | cow |
| bowl | cow | person | cup | bottle | cup |
| microwave | cup | book | dining table | apple | dining table |
| person | dining table | chair | dog | person | dog |
| laptop | dog | boat | elephant | person | elephant |
| bowl | elephant | dining table | fire hydrant | car | fire hydrant |
| airplane | fire hydrant | sandwich | fork | dining table | fork |
| car | fork | cup | giraffe | umbrella | giraffe |
| person | giraffe | cup | horse | person | horse |
| banana | horse | zebra | keyboard | truck | keyboard |
| mouse | keyboard | person | laptop | bottle | laptop |
| hair drier | motorcycle | book | motorcycle | person | motorcycle |
| giraffe | oven | sink | oven | cup | oven |
| laptop | person | car | person | dining table | pizza |
| person | pizza | cell phone | pizza | airplane | potted plant |
| person | potted plant | book | potted plant | dining table | refrigerator |
| microwave | refrigerator | oven | refrigerator | stop sign | sandwich |
| dining table | sandwich | dining table | sheep | person | sheep |
| orange | sheep | cat | sink | car | sink |
| bottle | sink | fork | suitcase | person | suitcase |
| bowl | surfboard | airplane | surfboard | person | surfboard |
| carrot | teddy bear | bowl | teddy bear | person | teddy bear |
| bottle | toilet | car | toilet | sink | toilet |
| cup | train | person | train | truck | train |
| dining table | truck | refrigerator | truck | person | truck |
| spoon | tv | chair | tv | car | tv |
| baseball bat | umbrella | person | umbrella | tv | zebra |
| giraffe | zebra | book | zebra | cardiomegaly | surgical clips |
| edema | chest tube | pleural effusion | chest tube | pneumothorax | chest tube |
| atelectasis | chest tube | cardiomegaly | chest tube | edema | endotracheal tube |
| pleural effusion | endotracheal tube | atelectasis | endotracheal tube | pneumothorax | endotracheal tube |
| cardiomegaly | endotracheal tube | edema | pacemaker | pleural effusion | pacemaker |
| pneumothorax | pacemaker | atelectasis | pacemaker | cardiomegaly | pacemaker |
| atelectasis | sternotomy wires | pneumothorax | sternotomy wires | cardiomegaly | sternotomy wires |
| edema | sternotomy wires | pleural effusion | sternotomy wires | edema | surgical clips |
| pleural effusion | surgical clips | atelectasis | surgical clips | pneumothorax | surgical clips |

In Figure 5, we summarize SYMBALBENCH with histograms detailing (1) the size of each dataset, (2) the strength of the injected systematic misalignment in each dataset as measured with Cramer's V, (3) the proportion of image-text pairs in each dataset containing the injected textual error $f$, and (4) the proportion of image-text pairs in each dataset containing the visual feature $g$. In Figure 6, we provide additional descriptive statistics on the natural image subset of SYMBALBENCH consisting of datasets derived from COCO; here, we provide histograms detailing (1) the mean size of the visual feature in each dataset (measured as proportion of total image area) and (2) the category of systematic misalignment (random, popular, or adversarial) as discussed in Appendix Section A.

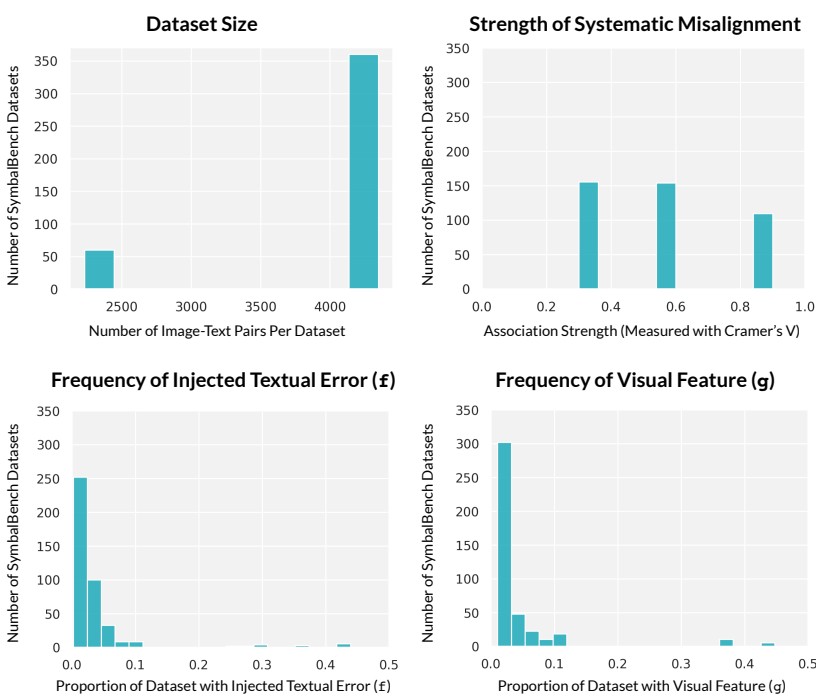

Figure 5: Here, we provide histograms summarizing the composition of datasets included in SYM-BALBENCH.

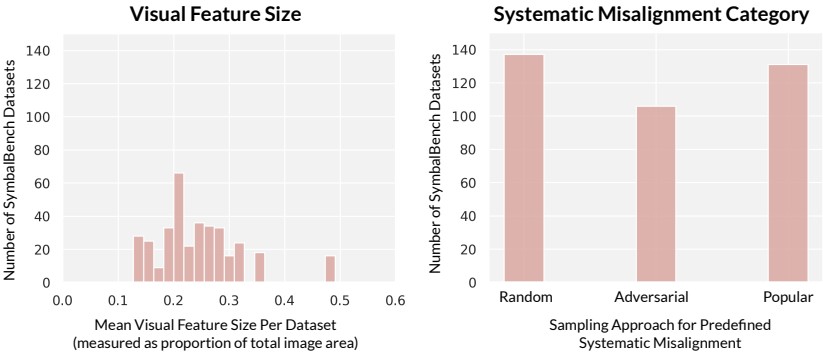

Figure 6: We provide additional descriptive statistics summarizing the composition of the 360 natural image datasets in SYMBALBENCH. We note here that if multiple sampling strategies yield the same predefined systematic misalignment, more than one category will be assigned to the same dataset; thus, the total count for the systematic misalignment category histogram may exceed 360.

## C  IMPLEMENTATION DETAILS FOR SYMBAL

SYMBAL decomposes the systematic misalignment detection task into two stages; here, we provide extended implementation details for each of these stages.

### C.1  IMPLEMENTATION DETAILS FOR SYMBAL STAGE 1

**Subtask 1: Grouping semantically-similar facts.** After aggregating all textual facts in $\mathcal{D}$ forming the set $\bigcup_{i=1}^{N} T_i$, we encode each fact using a text embedding model. For natural image datasets in SYMBALBENCH derived from COCO, we consider two options for text embedding models:

OpenCLIP-ViT-H-14-quickgelu (Ilharco et al., 2021) and Qwen3-Embedding-8B (Zhang et al., 2025). For medical image datasets in SYMBALBENCH derived from MIMIC-CXR, we consider three options for text embedding models: OpenCLIP-ViT-H-14-quickgelu (Ilharco et al., 2021), XRayCLIP-ViT-L (Chen et al., 2024b), and MedSigLIP (Sellergren et al., 2025). Of these, XrayCLIP-ViT-L and MedSigLIP are trained on radiology datasets. Embeddings are then clustered using spherical K-Means (implemented in Faiss (Johnson et al., 2019b)), where we sweep across a range of potential cluster numbers and select the optimal number of clusters using Silhouette distance; this approach is motivated by prior work (Sohoni et al., 2020; Varma et al., 2025).

**Subtask 2: Scoring groups by degree of misalignment.** We score each cluster by computing the average degree of alignment between constituent textual facts and paired images. We consider three possible scoring mechanisms, explained in detail below:

- *Embedding scorer:* Given a textual fact and its paired image, the embedding scorer utilizes an off-the-shelf vision-language model to compute embeddings for the text and image modalities. Alignment is measured by computing cosine similarity. This method is motivated by metrics like CLIPScore (Hessel et al., 2021), which have shown strong correlation with human judgments when measuring caption quality. For natural image datasets in SYMBALBENCH derived from COCO, we implement the embedding scorer with OpenCLIP-ViT-H-14-quickgelu (Ilharco et al., 2021) as the vision-language model. For medical image datasets in SYMBALBENCH derived from MIMIC-CXR, we consider three options for the embedding scorer: OpenCLIP-ViT-H-14-quickgelu (Ilharco et al., 2021), XRayCLIP-ViT-L (Chen et al., 2024b), and MedSigLIP (Sellergren et al., 2025). We note here that we do not use the embedding scorer in reference-based settings, since reference captions $C_i$ in our benchmark often have substantially more information than the single textual fact $f_k^i \in T_i$; this information imbalance is challenging to capture with embedding scorers.

- *Text-only scorer:* Given a textual fact and its paired image, the text-only scorer first generates a caption for the image and then prompts an LLM to determine if the textual fact is accurate with respect to the caption. For natural image datasets in SYMBALBENCH derived from COCO, we implement the text-only scorer using Llama-3.2-11B-Vision-Instruct (Grattafiori et al., 2024) to generate captions and Qwen2.5-VL-72B-Instruct (Qwen et al., 2025) to perform scoring. For medical image datasets in SYMBALBENCH derived from MIMIC-CXR, we implement the text-only scorer using Maira-2 (Bannur et al., 2024) to generate captions and Qwen2.5-VL-72B-Instruct (Qwen et al., 2025) or MedGemma-27B (Sellergren et al., 2025) to perform scoring. In the reference-based setting, we use the ground-truth caption $C_i$ rather than generating captions. We use the following input prompt in order to perform scoring:

> **Text-Only Scorer Input Prompt**
>
> You are provided with two image captions below, denoted as [A] and [B].
> [A]: <generated image caption or ground-truth reference caption>
> [B]: <candidate textual fact>
> Assume that [A] is the ground-truth caption. Is the content of [B] factually accurate with respect to [A]?
> Rules:
> 1. [B] may omit details from [A]; omission is acceptable.
> 2. If [B] introduces any incorrect or contradictory detail, it is inaccurate.
> Please output your answer as a single digit, where 1 indicates that [B] is accurate and 0 indicates that [B] is not accurate. Do not provide anything other than the digit in your response.

- *Vision-language scorer:* Given a textual fact and its paired image, the vision-language scorer provides an MLLM with both the image and the textual fact as input; the MLLM is then tasked with determining if the textual fact is accurate. For natural image datasets in SYMBALBENCH derived from COCO, we utilize Qwen2.5-VL-72B-Instruct (Qwen et al., 2025) as the MLLM. For medical image datasets in SYMBALBENCH derived from MIMIC-CXR, we utilize MedGemma-27B (Sellergren et al., 2025) as the MLLM. We use the following input prompt in the reference-free setting:

> **Vision-Language Scorer Input Prompt (Reference-Free)**
>
> <image>
> You are given an image. Below, a caption for the image is provided:
> Caption: <candidate textual fact>
> Is the caption accurate with respect to the image? Please output your answer as a single digit, where 1 indicates that the caption is accurate and 0 indicates that the caption is not accurate. Do not provide anything other than the digit in your response.

In the reference-based setting, we additionally provide the ground-truth reference caption to the MLLM. We use the following prompt in the reference-based setting:

> **Vision-Language Scorer Input Prompt (Reference-Based)**
>
> <image>
> You are provided an image as well as two image captions below, denoted as [A] and [B].
> [A]: <ground-truth reference caption>
> [B]: <candidate textual fact>
> Assume that [A] is the ground-truth caption. Is the content of [B] accurate with respect to the image? Please output your answer as a single digit, where 1 indicates that the caption is accurate and 0 indicates that the caption is not accurate. Do not provide anything other than the digit in your response.

**Subtask 3: Summarizing the top-ranked group.** We consider two summarization mechanisms for identifying the unifying concept shared by textual facts in $C_{text}$, discussed in detail below.

- *Embedding summarizer:* The embedding summarizer, which is used only for closed-ended settings, computes the cosine similarity between each textual fact in $C_{text}$ and the provided options. The cosine similarities are aggregated across all textual facts in $C_{text}$, and the option with the highest cosine similarity (or top-k highest cosine similarities) is selected as the output. For natural image datasets in SYMBALBENCH derived from COCO, we use OpenCLIP-ViT-H-14-quickgelu (Ilharco et al., 2021) to compute embeddings. For medical image datasets in SYMBALBENCH derived from MIMIC-CXR, we consider three possible models for generating embeddings: OpenCLIP-ViT-H-14-quickgelu (Ilharco et al., 2021), XRayCLIP-ViT-L (Chen et al., 2024b), and MedSigLIP (Sellergren et al., 2025).

- *Text-only summarizer:* The text-only summarizer provides an LLM with textual facts in $C_{text}$; the LLM is then tasked with identifying the unifying concept. For natural image datasets in SYMBALBENCH derived from COCO, we use Qwen2.5-VL-72B-Instruct (Qwen et al., 2025) as the LLM. For medical image datasets in SYMBALBENCH derived from MIMIC-CXR, we consider both Qwen2.5-VL-72B-Instruct (Qwen et al., 2025) and MedGemma-27B (Sellergren et al., 2025) as the LLM. In the closed-ended setting, we use the following input prompt. We then select the most frequently identified feature (or the top-k most frequently identified features) as output.

> **Text-Only Summarizer Input Prompt (Closed-Ended)**
>
> Consider this image caption: "<candidate textual fact>"
> From the following fixed list of options, identify the features that are present in the image.
> Options (you may only choose from these): <options>
> Output your answer in the following format:
> Answer: comma-separated list
> Rules:
> 1. The caption may use different words to describe features. Treat any visually equivalent description as matching an option.
> 2. Do NOT include any text outside the options above.
> 3. Do NOT explain your reasoning.
> 4. If none of the features are present, output an empty list of the form: "Answer: "

In the open-ended setting, we use the following input prompt. Then, given the output, we prompt the same LLM to select the most frequently identified feature (or the top-k most frequently identified features) as output.

---

**Text-Only Summarizer Input Prompt (Open-Ended)**

Consider this image caption: "<candidate textual fact>"
Identify the visual features that are present in the image.
Output your answer in the following format:
Answer: comma-separated list

Rules:
1. Each feature should be described concisely in a single phrase.
2. Each feature must be directly visible in the image.
3. Do NOT include any text outside the identified features.
4. Do NOT explain your reasoning.
5. If no features are present, output an empty list of the form: "Answer: "

---

## C.2 IMPLEMENTATION DETAILS FOR SYMBAL STAGE 2

**Subtask 1: Grouping semantically-similar images.** For natural image datasets in SYMBALBENCH derived from COCO, we consider two options for image embedding models: OpenCLIP-ViT-H-14-quickgelu (Ilharco et al., 2021) and DINOv2-ViT-L-14 (Oquab et al., 2024). For medical image datasets in SYMBALBENCH derived from MIMIC-CXR, we consider three options for image embedding models: OpenCLIP-ViT-H-14-quickgelu (Ilharco et al., 2021), XRayCLIP-ViT-L (Chen et al., 2024b), and MedSigLIP (Sellergren et al., 2025). Similar to Stage 1, embeddings are clustered using spherical K-Means, where we sweep across a range of potential cluster numbers and select the optimal number of clusters using Silhouette distance.

**Subtask 2: Scoring groups by degree of misalignment.** We score each cluster by computing the mean degree of misalignment between images and paired textual facts in $C_{text}$. We consider the same scoring mechanisms as in Stage 1.

**Subtask 3: Summarizing the top-ranked group.** We consider three summarization mechanisms for identifying the unifying concept shared by images in $C_{image}$, described in detail below.

- *Embedding summarizer:* The embedding summarizer, which is used only for closed-ended settings, computes the cosine similarity between each image in $C_{image}$ and the provided options. The cosine similarities are aggregated across all images in $C_{image}$, and the option with the highest cosine similarity (or top-k highest cosine similarities) is selected as the output. For natural image datasets in SYMBALBENCH derived from COCO, we use OpenCLIP-ViT-H-14-quickgelu (Ilharco et al., 2021) to compute embeddings. For medical image datasets in SYMBALBENCH derived from MIMIC-CXR, we consider three possible models for generating embeddings: OpenCLIP-ViT-H-14-quickgelu (Ilharco et al., 2021), XRayCLIP-ViT-L (Chen et al., 2024b), and MedSigLIP (Sellergren et al., 2025).

- *Text-only summarizer:* The text-only summarizer generates a caption for each image in $C_{image}$; then, an LLM is tasked with identifying the unifying concept. For natural image datasets in SYMBALBENCH derived from COCO, captions are generated using Llama-3.2–1B-Vision-Instruct Grattafiori et al. (2024). For medical image datasets in SYMBALBENCH, captions are generated using MAIRA-2 Bannur et al. (2024). In reference-based settings, we use the ground-truth reference captions rather than generating captions. We use the same prompts and models as discussed above in Stage 1, Subtask 3.

- *Vision-language summarizer:* The vision-language summarizer provides an MLLM with images in $C_{image}$; then, the MLLM is prompted to identify the unifying concept. For natural image datasets in SYMBALBENCH derived from COCO, we use Qwen2.5-VL-72B-Instruct (Qwen et al., 2025) as the MLLM. For medical image datasets in SYMBALBENCH derived from MIMIC-CXR, we use MedGemma-27B (Sellergren et al., 2025) as the MLLM. For reference-based settings, we also provide the ground-truth reference caption to the MLLM. In the closed-ended setting, we use the

following input prompt. We then select the most frequently-identified feature (or the top-k most frequently identified features) as output:

---

**Vision-Language Summarizer Input Prompt (Closed-Ended)**

<image>
Consider this image.
From the following fixed list of options, identify the features that are present in the image.
Options (you may only choose from these): <options>
Output your answer in the following format:
Answer: comma-separated list
Rules:
1. Do NOT include any text outside the options above.
2. Do NOT explain your reasoning.
3. If none of the features are present, output an empty list of the form: "Answer: "

---

In the open-ended setting, we use the following input prompt. Then, given the outputs, we prompt the same MLLM to select the most frequently identified feature (or the top-k most frequently identified features) as output.

---

**Vision-Language Summarizer Input Prompt (Open-Ended)**

<image>
Consider this image.

Identify the visual features that are present in the image.
Output your answer in the following format:
Answer: comma-separated list
Rules:
1. Each feature should be described concisely in a single phrase.
2. Each feature must be directly visible in the image.
3. Do NOT include any text outside the identified features.
4. Do NOT explain your reasoning.
5. If no features are present, output an empty list of the form: "Answer: "
6. Include a maximum of ten features.

---

## D    EXTENDED RESULTS

In Table 4, we provide an extended version of Table 1, extending to the top-ten compositions. Note that Table 4 only includes compositions that can support all four SYMBALBENCH experimental settings; for instance, embedding-based summarizers are excluded from this table due to lack of support on open-ended settings.

In Table 5, we provide an extended version of Table 2, extending to the top-ten compositions. Again, Table 5 only includes compositions that can support all four SYMBALBENCH experimental settings.

In Table 6, we provide a tabular version of Figure 3 stratified by domain.

We use the following input prompt for our direct-prompting baselines in the open-ended setting:

---

**Direct-Prompting Baseline Input Prompt (Open-Ended)**

You are provided with a dataset, where each sample consists of the following two components:

Reference caption: A ground-truth caption describing the content of an image
Model-generated caption: A caption generated by an AI model

The model-generated captions may have systematic errors, where a recurring textual error is closely associated with the presence of a specific visual feature in the paired image. Your task is to identify the recurring textual error and the associated visual feature.

Output your answer in the following format, where each comma-separated list consists of your top-five predictions in order:
Textual Error: comma-separated list
Visual Feature: comma-separated list

Rules:
1. Each visual feature must be directly visible in the image.
2. Do NOT include any text outside of the answer.
3. Do NOT explain your reasoning.

Dataset: <samples from dataset with images expressed in text-form>

---

We use the following input prompt for our direct-prompting baselines in the closed-ended setting:

---

**Direct-Prompting Baseline Input Prompt (Closed-Ended)**

You are provided with a dataset, where each sample consists of the following two components:

Reference caption: A ground-truth caption describing the content of an image
Model-generated caption: A caption generated by an AI model

The model-generated captions may have systematic errors, where a recurring textual error is closely associated with the presence of a specific visual feature in the paired image. Your task is to identify the recurring textual error and the associated visual feature.

Output your answer in the following format, where each comma-separated list consists of your top-five predictions in order:
Textual Error: comma-separated list
Visual Feature: comma-separated list

Select the textual error from the following list of options (you may only choose from these):
<textual choices>
Select the visual feature from the following list of options (you may only choose from these):
<visual choices>

Rules:
1. Do NOT include any text outside of the options above.
2. Do NOT explain your reasoning.

Dataset: <samples from dataset with images expressed in text-form>

---

In Figure 7, we extend Figure 4 by providing a breakdown of SYMBAL performance across various categories of systematic misalignments in the natural image subset of SYMBALBENCH.

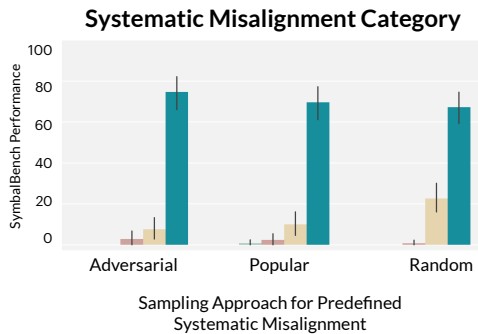

Figure 7: We provide a breakdown of SYMBAL performance across various categories of systematic misalignments in the natural image subset of SYMBALBENCH.

Table 4: We consider the role of various text embedding models, alignment scorers, and summarizers on the performance of Stage 1 of SYMBAL. Here, VL refers to the vision-language scorer and MG-27B refers to MedGemma-27B.

| | Text Embedding | Alignment Scorer | Summarizer | Reference-Free | | | | Reference-Based | | | |
| | | | | Closed-Ended | | Open-Ended | | Closed-Ended | | Open-Ended | |
| | | | | Acc@1 | Acc@5 | Acc@1 | Acc@5 | Acc@1 | Acc@5 | Acc@1 | Acc@5 |
|---|---|---|---|---|---|---|---|---|---|---|---|
| Natural | Qwen3-8B | VL (Qwen-72B) | Text (Qwen-72B) | **93.9** | **94.4** | **92.8** | **94.2** | 84.4 | 85.3 | 80.8 | 82.8 |
| | OpenCLIP | VL (Qwen-72B) | Text (Qwen-72B) | **93.9** | **94.4** | **92.8** | 93.9 | 87.2 | 88.6 | 86.1 | 87.8 |
| | Qwen3-8B | Text (Qwen-72B) | Text (Qwen-72B) | 83.9 | 85.8 | 82.8 | 85.0 | 84.2 | 85.3 | 81.9 | 83.9 |
| | OpenCLIP | Text (Qwen-72B) | Text (Qwen-72B) | 66.1 | 68.6 | 64.2 | 67.2 | 70.6 | 72.2 | 67.5 | 71.4 |
| Medical | XRayCLIP | Text (MG-27B) | Text (MG-27B) | **58.3** | – | **51.7** | **75.0** | **100.0** | – | 88.3 | 95.0 |
| | XRayCLIP | Text (MG-27B) | Text (Qwen-72B) | 56.7 | – | **51.7** | 73.3 | **100.0** | – | **100.0** | **100.0** |
| | XRayCLIP | Text (Qwen-72B) | Text (MG-27B) | 31.7 | – | 26.7 | 58.3 | 98.3 | – | 90.0 | 93.3 |
| | MedSigLIP | Text (MG-27B) | Text (MG-27B) | 45.0 | – | 30.0 | 53.3 | **100.0** | – | 83.3 | **100.0** |
| | XRayCLIP | VL (MG-27B) | Text (MG-27B) | 36.7 | – | 26.7 | 48.3 | 93.3 | – | 85.0 | 90.0 |
| | XRayCLIP | Text (Qwen-72B) | Text (Qwen-72B) | 33.3 | – | 28.3 | 46.7 | **100.0** | – | 98.3 | 98.3 |
| | OpenCLIP | Text (MG-27B) | Text (MG-27B) | 43.3 | – | 28.3 | 46.7 | **100.0** | – | 88.3 | 98.3 |
| | OpenCLIP | Text (MG-27B) | Text (Qwen-72B) | 41.7 | – | 36.7 | 45.0 | **100.0** | – | 98.3 | **100.0** |
| | MedSigLIP | Text (MG-27B) | Text (Qwen-72B) | 43.3 | – | 36.7 | 43.3 | **100.0** | – | 98.3 | **100.0** |
| | MedSigLIP | Text (Qwen-72B) | Text (MG-27B) | 28.3 | – | 16.7 | 35.0 | **100.0** | – | 86.7 | 98.3 |

# E    APPLYING SYMBAL TO REAL-WORLD SETTINGS

In this section, we further demonstrate the utility of SYMBAL by supplementing our evaluations on SYMBALBENCH with additional quantitative and qualitative analyses in real-world settings.

**SYMBAL can accurately surface systematic misalignments in captions generated by off-the-shelf MLLMs.** Below, we list several examples of systematic misalignments identified by SYMBAL, and we also provide associated validation:

- *Example 1:* In captions generated by Llava1.5-7B, SYMBAL detects that erroneous references to a TV ($\hat{f}$) in captions are often systematically associated with the presence of a desk, computer monitor, and/or keyboard ($\hat{g}$) in the scene. We provide visual examples of image-caption pairs with the SYMBAL-identified systematic misalignment in Figure 8 (Row 1). Quantitatively, our analysis finds that erroneous references to a TV in model-generated captions are indeed 13.5 times more likely when a desk is present in the image compared to when a desk is absent, validating the SYMBAL prediction.

- *Example 2:* In captions generated by Llava1.5-7B, SYMBAL detects that erroneous references to a handbag or a handbag on the ground ($\hat{f}$) in captions are often systematically associated with the presence of a bus ($\hat{g}$) in a scene. We provide visual examples of image-caption pairs with the SYMBAL-identified systematic misalignment in Figure 8 (Row 2). Quantitatively, our analysis finds that erroneous references to a handbag in model-generated captions are indeed 3.1 times more likely when a bus is present in the image compared to when a bus is absent, validating the SYMBAL prediction.

- *Example 3:* In captions generated by Llava1.5-7B, SYMBAL detects that erroneous references to a chair ($\hat{f}$) in captions are often systematically associated with the presence of a television ($\hat{g}$) in

Table 5: We consider the role of various image embedding models, alignment scorers, and summarizers on the performance of Stage 2 of SYMBAL. Here, VL refers to the vision-language scorer, Emb. refers to the embedding scorer, and MG-27B refers to MedGemma-27B.

| | Img Embedding | Alignment Scorer | Summarizer | Reference-Free | | | | Reference-Based | | | |
| | | | | Closed-Ended | | Open-Ended | | Closed-Ended | | Open-Ended | |
| | | | | Acc@1 | Acc@5 | Acc@1 | Acc@5 | Acc@1 | Acc@5 | Acc@1 | Acc@5 |
|---|---|---|---|---|---|---|---|---|---|---|---|
| Natural | OpenCLIP | VL (Qwen-72B) | Text (Qwen-72B) | 52.2 | 71.1 | **49.7** | **69.7** | 42.5 | 60.3 | 41.9 | 52.2 |
| | OpenCLIP | Emb. (OpenCLIP) | VL (Qwen-72B) | 53.9 | 71.4 | 48.1 | 63.9 | 45.6 | 57.8 | 42.5 | 55.6 |
| | OpenCLIP | Emb. (OpenCLIP) | Text (Qwen-72B) | 48.6 | 67.5 | 47.8 | 62.8 | 45.3 | 59.2 | 43.9 | 55.8 |
| | OpenCLIP | VL (Qwen-72B) | VL (Qwen-72B) | 53.9 | 70.6 | 45.8 | 62.5 | 44.4 | 59.2 | 38.9 | 52.2 |
| | DINOv2 | VL (Qwen-72B) | Text (Qwen-72B) | 46.9 | 64.2 | 45.3 | 61.4 | 41.4 | 57.5 | 38.6 | 54.7 |
| | DINOv2 | Text (Qwen-72B) | Text (Qwen-72B) | 45.6 | 65.0 | 43.1 | 60.8 | 40.3 | 58.1 | 41.1 | 56.4 |
| | OpenCLIP | Text (Qwen-72B) | Text (Qwen-72B) | 51.9 | 69.2 | 48.1 | 60.6 | 46.4 | 59.2 | **45.6** | 58.1 |
| | OpenCLIP | Text (Qwen-72B) | VL (Qwen-72B) | **55.0** | **72.8** | 44.2 | 60.3 | **48.1** | **62.2** | 43.9 | **56.7** |
| | DINOv2 | Text (Qwen-72B) | VL (Qwen-72B) | 48.6 | 70.6 | 43.6 | 59.7 | 46.7 | 61.4 | 39.7 | 54.2 |
| | DINOv2 | Embedding (OpenCLIP) | VL (Qwen-72B) | 54.2 | 69.2 | 43.6 | 59.4 | 47.2 | 60.8 | 39.7 | 53.3 |
| Medical | XRayCLIP | Emb. (MedSigLIP) | VL (MG-27B) | **26.7** | – | 11.7 | **36.7** | 41.7 | – | 28.3 | 53.3 |
| | MedSigLIP | Emb. (MedSigLIP) | VL (MG-27B) | 23.3 | – | 11.7 | 31.7 | 40.0 | – | 25.0 | 46.7 |
| | OpenCLIP | Emb. (MedSigLIP) | VL (MG-27B) | 23.3 | – | 13.3 | 28.3 | 35.0 | – | 20.0 | 46.7 |
| | MedSigLIP | Emb. (XRayCLIP) | VL (MG-27B) | 25.0 | – | 10.0 | 28.3 | 50.0 | – | 33.3 | 60.0 |
| | XRayCLIP | VL (MG-27B) | VL (MG-27B) | 21.7 | – | 6.7 | 28.3 | **61.7** | – | **43.3** | **65.0** |
| | MedSigLIP | Text (MG-27B) | VL (MG-27B) | 25.0 | – | 8.3 | 26.7 | **61.7** | – | **43.3** | **65.0** |
| | OpenCLIP | Text (MG-27B) | VL (MG-27B) | 13.3 | – | 10.0 | 25.0 | 48.3 | – | 23.3 | 63.3 |
| | OpenCLIP | Text (Qwen-72B) | VL (MG-27B) | 21.7 | – | 3.3 | 25.0 | 46.7 | – | 30.0 | 61.7 |
| | MedSigLIP | Embedding (MedSigLIP) | Text (Qwen-72B) | 15.0 | – | **15.0** | 25.0 | 46.7 | – | 15.0 | 40.0 |
| | OpenCLIP | Embedding (MedSigLIP) | Text (Qwen-72B) | 18.3 | – | 13.3 | 23.3 | 46.7 | – | 16.7 | 48.3 |

Table 6: End-to-end performance across SYMBALBENCH, stratified by domain.

| | Method | Reference-Free | | | | Reference-Based | | | |
| | | Closed-Ended | | Open-Ended | | Closed-Ended | | Open-Ended | |
| | | Acc@1 | Acc@5 | Acc@1 | Acc@5 | Acc@1 | Acc@5 | Acc@1 | Acc@5 |
|---|---|---|---|---|---|---|---|---|---|
| Natural | Random | 0.0 | 0.6 | – | – | 0.0 | 0.6 | – | – |
| | Llama3.3-70B | 0.0 | 1.7 | 0.3 | 0.3 | 0.6 | 2.5 | 0.6 | 1.4 |
| | Qwen2.5-VL-72B | 1.4 | 2.8 | 0.0 | 1.9 | 1.7 | 3.3 | 0.6 | 1.1 |
| | GPT-OSS 120B | 11.7 | 16.4 | 9.2 | 13.9 | 16.1 | 21.4 | 10.8 | 17.2 |
| | SYMBAL (Ours) | **52.2** | **71.1** | **49.2** | **69.7** | **42.5** | **60.3** | **41.1** | **51.9** |
| Medical | Random | 3.3 | – | – | – | 3.3 | – | – | – |
| | Llama3.3-70B | 3.3 | – | 0.0 | 8.3 | 5.0 | – | 0.0 | 5.0 |
| | MedGemma-27B | 10.0 | – | 0.0 | 1.7 | 5.0 | – | 0.0 | 0.0 |
| | Qwen2.5-VL-72B | 10.0 | – | 3.3 | 5.0 | 13.3 | – | 0.0 | 1.7 |
| | GPT-OSS 120B | 6.7 | – | 1.7 | 21.7 | 23.3 | – | 0.0 | 11.7 |
| | SYMBAL (Ours) | **18.3** | – | **6.7** | **28.3** | **41.7** | – | **25.0** | **48.3** |

a scene. We provide visual examples of image-caption pairs with the SYMBAL-identified systematic misalignment in Figure 8 (Row 3). Quantitatively, our analysis finds that erroneous references to a `chair` in model-generated captions are indeed 3.1 times more likely when a `television` is present in the image compared to when a `television` is absent, validating the SYMBAL prediction.

- *Example 4:* In captions generated by Llava1.5-13B, SYMBAL detects that erroneous references to a `TV` ($\hat{f}$) in captions are often systematically associated with the presence of a `computer monitor`, `keyboard`, and/or `mouse` ($\hat{g}$) in a scene. Interestingly, this systematic misalignment is nearly identical to one that exists in Llava1.5-7B-generated captions (see Example 1), suggesting that solely increasing the scale of the underlying MLLM is insufficient for resolving systematic misalignments. We provide visual examples of image-caption pairs with the SYMBAL-identified systematic misalignment in Figure 9 (Row 1). Quantitatively, our analysis finds that erroneous references to a `TV` in model-generated captions are indeed 22.2 times more likely when a `computer monitor` is present in the image compared to when a `computer monitor` is absent, validating the SYMBAL prediction.

- *Example 5:* In captions generated by LlavaOneVision-7B, SYMBAL detects that erroneous references to `text` ($\hat{f}$) in captions are often systematically associated with the presence of a `sign` ($\hat{g}$) in a scene. This systematic misalignment suggests that LlavaOneVision-7B struggles with OCR capabilities, where the presence of text-based signage in an image is likely to result in errors in the generated caption. We provide visual examples of image-caption pairs with the SYMBAL-identified systematic misalignment in Figure 9 (Row 2). Quantitatively, our analysis finds that erroneous references to `text` in model-generated captions are indeed 4.6 times more likely when a `sign` is present in the image compared to when a `sign` is absent, validating the SYMBAL prediction.

- *Example 6:* In captions generated by AyaVision-8B, SYMBAL detects that erroneous references to a vase ($\hat{f}$) in captions are often systematically associated with the presence of a couch ($\hat{g}$) in a scene. We provide visual examples of image-caption pairs with the SYMBAL-identified systematic misalignment in Figure 9 (Row 3). Quantitatively, our analysis finds that erroneous references to a vase in model-generated captions are indeed 17.7 times more likely when a couch is present in the image compared to when a couch is absent, validating the SYMBAL prediction.

Across all six examples of SYMBAL-identified systematic misalignments provided above, we find that erroneous references to $\hat{f}$ are substantially more likely when $\hat{g}$ is present in the image compared to when $\hat{g}$ is absent. This analysis validates discovered misalignments by demonstrating that links between SYMBAL-identified erroneous textual fact $\hat{f}$ and SYMBAL-identified visual feature $\hat{g}$ do indeed exist.

Our quantitative validation procedure relies on automated annotation methods in order to enable evaluation at scale; in particular, we leverage Qwen-72B in order to annotate erroneous references to $\hat{f}$ in each caption. We find that these generated annotations align closely with human judgments. Given the set of 215 images in the dataset containing a "bus", we tasked a human reader with identifying whether each Llava1.5-7B-generated caption contained an erroneous reference to a "handbag" and/or "handbag on the ground" (Example 2). Human judgments aligned perfectly with Qwen-72B predictions in 96.3% of cases (Cohen's kappa = 0.86).

**SYMBAL is a powerful tool for auditing open-source vision-language datasets.** Below, we list several examples of systematic misalignments identified by SYMBAL on the ShareGPT4V dataset, and we also provide associated validation:

- *Example 7:* SYMBAL detects that erroneous references to a white tablecloth ($\hat{f}$) in captions are often systematically associated with the presence of a table, cake, and/or people ($\hat{g}$) in the scene. We provide visual examples of image-caption pairs with the SYMBAL-identified systematic misalignment in Figure 10 (Row 1). Quantitatively, our analysis finds that erroneous references to a white tablecloth in model-generated captions are indeed 17.2 times more likely when a table is present in the image compared to when a table is absent, validating the SYMBAL prediction.

- *Example 8:* SYMBAL detects that erroneous references to a printer ($\hat{f}$) in captions are often systematically associated with the presence of a computer monitor ($\hat{g}$) in a scene. We provide visual examples of image-caption pairs with the SYMBAL-identified systematic misalignment in Figure 10 (Row 2). Quantitatively, our analysis finds that erroneous references to a printer in model-generated captions are indeed 121 times more likely when a computer monitor is present in the image compared to when a computer monitor is absent, validating the SYMBAL prediction.

- *Example 9:* SYMBAL detects that erroneous references to a black phone ($\hat{f}$) in captions are often systematically associated with the presence of a laptop ($\hat{g}$) in a scene. We provide visual examples of image-caption pairs with the SYMBAL-identified systematic misalignment in Figure 10 (Row 3). Quantitatively, our analysis finds that erroneous references to a black phone in model-generated captions are indeed 48.5 times more likely when a laptop is present in the image compared to when a laptop is absent, validating the SYMBAL prediction.

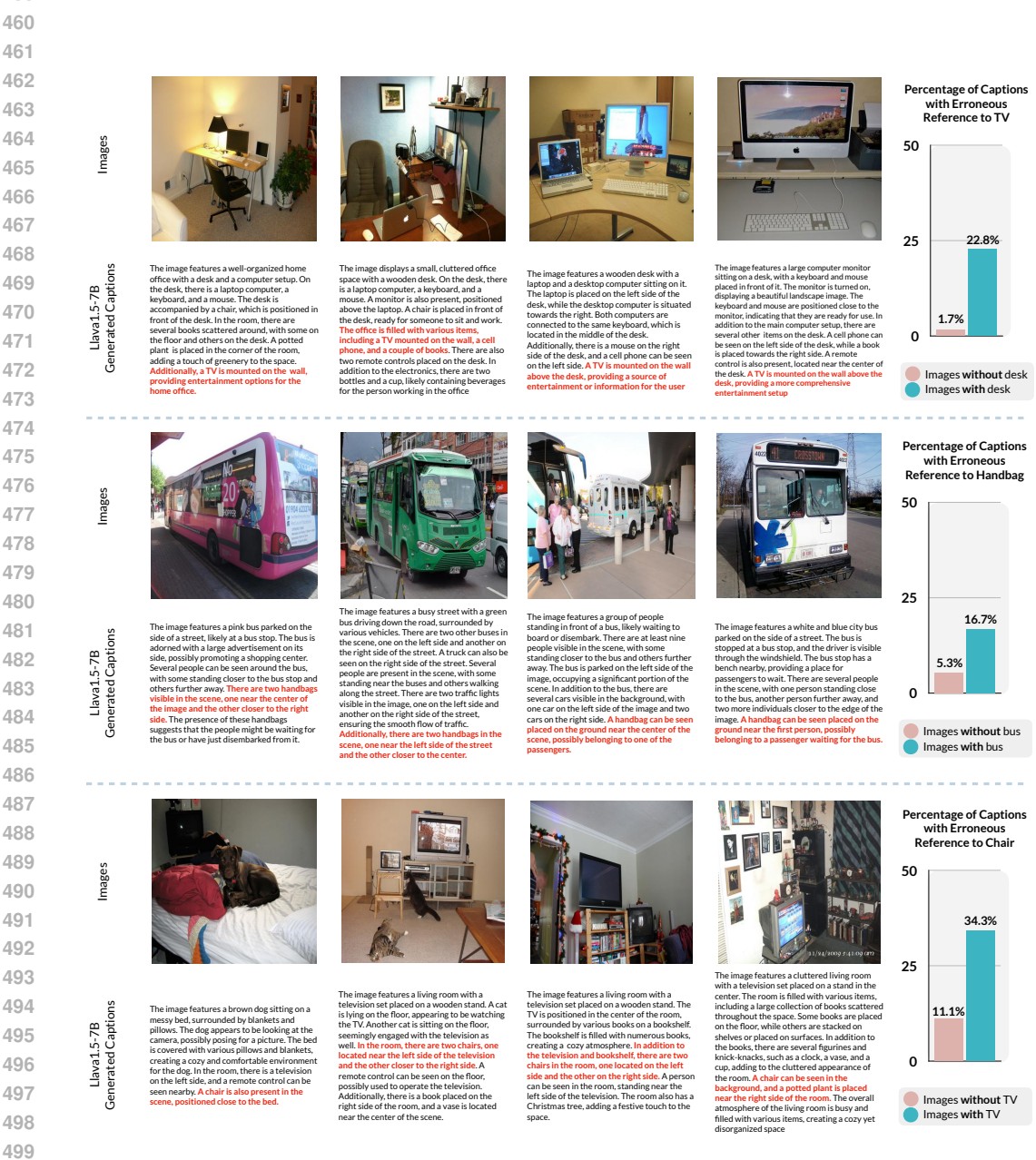

Figure 8: Examples of image-caption pairs with SYMBAL-identified systematic misalignments are shown here, with the identified erroneous textual fact in each caption highlighted in red. We also quantitatively validate each identified systematic misalignment. [Row 1] SYMBAL detects that erroneous references to a TV ($\hat{f}$) in captions are often systematically associated with the presence of a desk, computer monitor, and/or keyboard ($\hat{g}$) in the scene. [Row 2] SYMBAL detects that erroneous references to a handbag or handbag on the ground($\hat{f}$) in captions are often systematically associated with the presence of a bus ($\hat{g}$) in a scene. [Row 3] SYMBAL detects that erroneous references to a chair ($\hat{f}$) in captions are often systematically associated with the presence of a television ($\hat{g}$) in a scene.

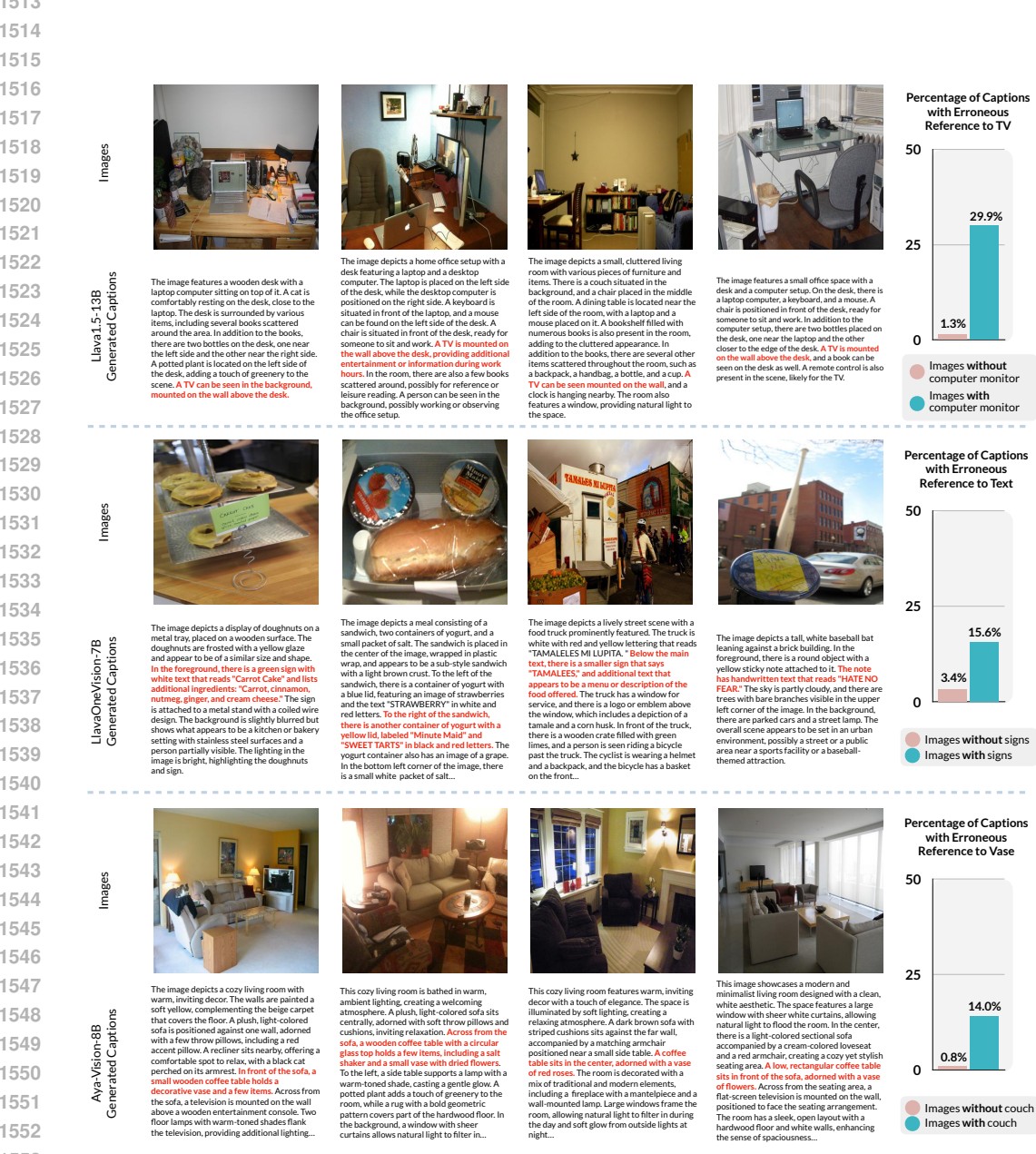

Figure 9: Examples of image-caption pairs with SYMBAL-identified systematic misalignments are shown here, with the identified erroneous textual fact in each caption highlighted in red. We also quantitatively validate each identified systematic misalignment. [Row 1] SYMBAL detects that erroneous references to a TV ($\hat{f}$) in Llava1.5-13B-generated captions are often systematically associated with the presence of a computer monitor, keyboard, and/or mouse ($\hat{g}$) in the scene. [Row 2] SYMBAL detects that erroneous references to text ($\hat{f}$) in LlavaOneVision-7B-generated captions are often systematically associated with the presence of a sign ($\hat{g}$) in a scene. [Row 3] SYMBAL detects that erroneous references to a vase ($\hat{f}$) in AyaVision-8B-generated captions are often systematically associated with the presence of a couch ($\hat{g}$) in a scene.

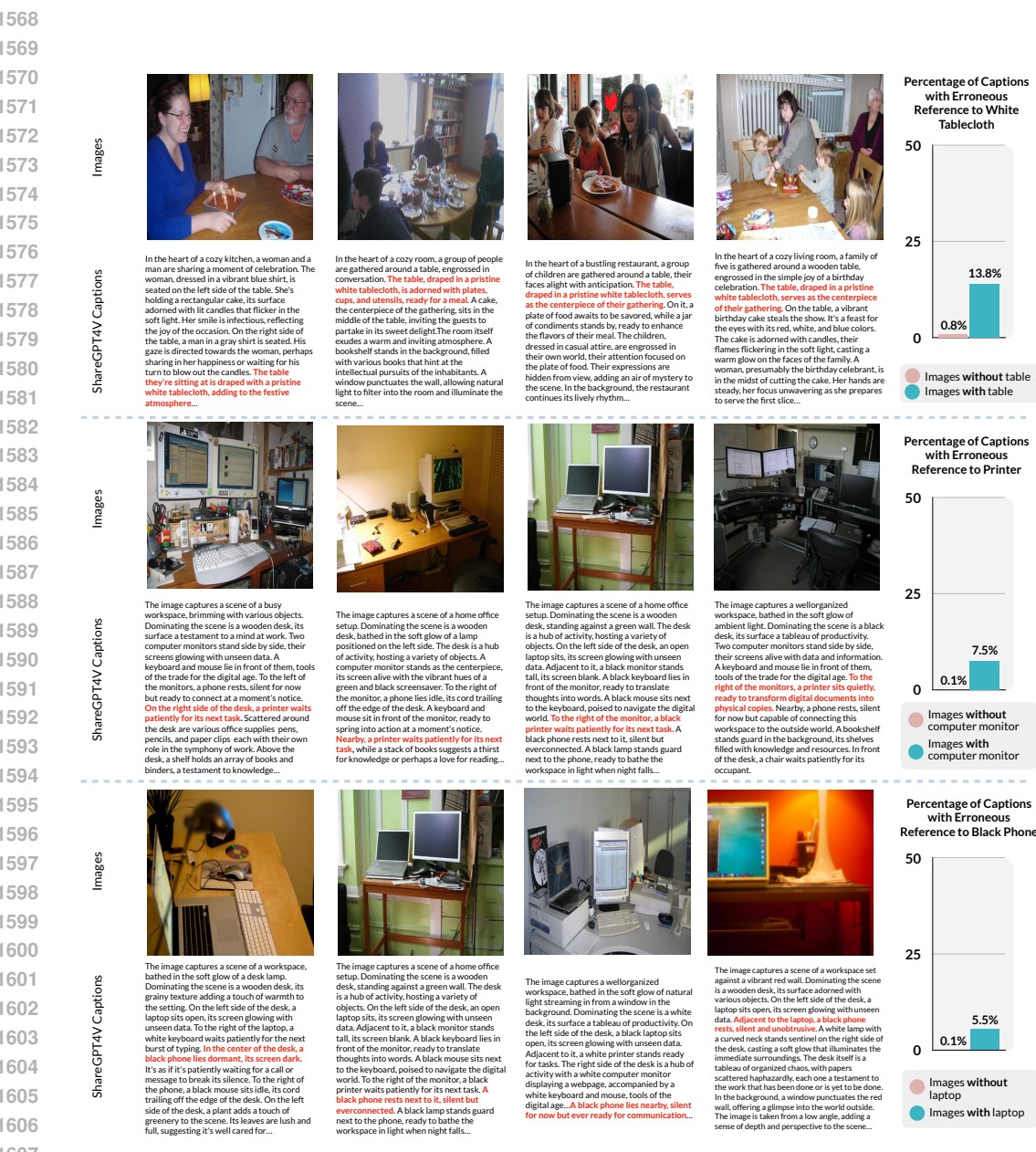

Figure 10: Examples of image-caption pairs with SYMBAL-identified systematic misalignments are shown here, with the identified erroneous textual fact in each caption highlighted in red. We also quantitatively validate each identified systematic misalignment. [Row 1] SYMBAL detects that erroneous references to a white tablecloth ($\hat{f}$) in ShareGPT4V captions are often systematically associated with the presence of a table, cake, and/or people ($\hat{g}$) in the scene. [Row 2] SYMBAL detects that erroneous references to a printer ($\hat{f}$) in ShareGPT4V captions are often systematically associated with the presence of a computer monitor ($\hat{g}$) in a scene. [Row 3] SYMBAL detects that erroneous references to a black phone ($\hat{f}$) in ShareGPT4V captions are often systematically associated with the presence of a laptop ($\hat{g}$) in a scene.

