# OpenReview forum: "Symbal: Detecting Systematic Misalignments in Model-Generated Captions"
_ICLR.cc/2026/Conference — Submitted to ICLR 2026_

### Official Review · Reviewer_LmQg · 2025-10-30

**Soundness:** 3
**Presentation:** 3
**Contribution:** 3
**Rating:** 8
**Confidence:** 3

**Summary:**

The paper introduces a new task called systematic misalignment detection, which aims to find recurring caption errors in MLLM-generated image captions that are systematically tied to specific visual features (e.g., “cardiomegaly” errors co-occurring with “pacemaker” presence). It contributes (1) **SYMBALBENCH**, a benchmark of 420 synthetic vision-language datasets with planted (f, g) misalignments spanning natural and medical images, and (2) **SYMBAL**, a two-stage pipeline that first discovers the erroneous textual fact ( $\hat f$ ) from captions and then the associated visual feature ( $\hat g$ ) from images, each via clustering, alignment scoring, and summarization. The benchmark is created by injecting textual facts with controlled association strength (Cramer’s V) to selected visual features; evaluation uses Accuracy@K and, for open-ended outputs, LLM-as-a-Judge.

**Strengths:**

1. The two-stage structure is intuitive and defensible, with transparent sub-steps (grouping/score/summarize). This factorization plausibly explains gains over single-shot prompting.
2. The paper compares multiple text/image embedding backbones and alignment scorers for each stage and reports Accuracy@1/5 across reference-free and reference-based variants, highlighting where VL scorers help and where medical-domain models are needed.
3. Appendix examples suggest the system can surface plausible spurious ties in off-the-shelf MLLM captions.
4. The problem identified in this paper holds significant value, and the method solving it is well motivated.

**Weaknesses:**

1. Although the benchmark is large and controllable, its errors are *injected*. This risks overfitting to the injection mechanism or distributional artifacts.
2. Visual-feature detection in medical images remains modest, which could constrain practical utility in safety-critical use cases where the visual cue is subtle or device-like features are variable.
3. Add analyses where (a) f and g are *correlated but not causal*, (b) g is *tiny/occluded*, and (c) multiple candidate g’s co-occur, to probe failure modes beyond Cramer’s-V control.
4. Stage-wise pipelines use powerful VL models (e.g., Qwen2.5-72B) both to *score alignment* and to *summarize clusters*. If those models encode similar biases to the caption generator, they might either mask or hallucinate misalignments. Controls isolating scorer influence (e.g., cross-model scoring) are only partially addressed.

**Questions:**

N/A

---

> ### Author Response · Authors · 2025-11-26
> **Response to Reviewer LMQG [1/2]**
>
> We thank Reviewer LMQG for reviewing our work and for providing helpful feedback.
>
> > **[Q1] Benchmark composition. “Although the benchmark is large and controllable, its errors are injected. This risks overfitting to the injection mechanism or distributional artifacts.”**
>
> We refer the reviewer to General Response [Q1], where we address this point.
>
> > **[Q2] Performance on medical settings. “Visual-feature detection in medical images remains modest, which could constrain practical utility in safety-critical use cases where the visual cue is subtle or device-like features are variable.”**
>
> We agree with the reviewer that datasets from the medical domain are challenging, and we identify two contributing factors. First, as medical images are more visually complex than natural images, subtasks like alignment scoring and summarization are inherently more difficult in the medical setting. Second, research into domain-specific medical LLMs and MLLMs has been limited in comparison to general-domain counterparts, resulting in observed performance disparities across domains.
>
> Our results affirm that SymbalBench is a challenging and practically-useful benchmark. We hope that SymbalBench will inspire the development of future methods for addressing the systematic misalignment detection task.
>
> > **[Q3] Additional analyses. “Add analyses where (a) f and g are correlated but not causal, (b) g is tiny/occluded, and (c) multiple candidate g’s co-occur, to probe failure modes beyond Cramer’s-V control.”**
>
> Thank you for this suggestion. Below, we provide details related to each of these three lines of analyses.
> - **Analyses where the erroneous textual fact and associated visual feature are correlated but not causal:** All SymbalBench datasets are designed such that the erroneous textual fact $f$ and associated visual feature $g$ are correlated but not causally linked. Thus, all quantitative evaluations in Section 5.1, 5.2, and 5.3 fall under this category of analyses. We refer the reviewer to Appendix Section A as well as our response to [Q1] above, where we provide additional methodological details on selecting $f$ and $g$ such that this property is satisfied.
> - **Analyses where the visual feature is tiny:** In real-world datasets, visual features exhibit varying sizes, where smaller features present a particularly challenging setting. As shown in Figure 6, SymbalBench encompasses visual features of diverse sizes. In Figure 4, we report performance of Symbal stratified across visual feature sizes, where we demonstrate that Symbal outperforms baselines on challenging SymbalBench datasets where visual features $g$ are small.
> - **Analyses where multiple candidate misalignments co-occur:** Real-world datasets are likely to include multiple systematic misalignments. In order to empirically demonstrate that Symbal is capable of operating in this setting, we have updated our manuscript with substantial extensions to our real-world evaluations. In particular, we show that Symbal can accurately detect multiple real-world systematic misalignments in captions generated by Llava1.5-7B; in Figure 8, we demonstrate with both qualitative and quantitative analyses that these Symbal-predicted misalignments are accurate. We also show that Symbal can accurately detect multiple systematic misalignments in ShareGPT4V, an off-the-shelf dataset with MLLM-generated captions (Figure 10). We refer the reviewer to Section 6.4, Appendix Section E, Figure 8, and Figure 10 of our updated manuscript, where these extended evaluations are described in detail.
>
> [Continued in next comment]

---

> ### Author Response · Authors · 2025-11-26
> **Response to Reviewer LMQG [2/2]**
>
> > **[Q4] Use of off-the-shelf VL models. “Stage-wise pipelines use powerful VL models (e.g., Qwen2.5-72B) both to score alignment and to summarize clusters. If those models encode similar biases to the caption generator, they might either mask or hallucinate misalignments. Controls isolating scorer influence (e.g., cross-model scoring) are only partially addressed.”**
>
> Thank you for raising this point. While it is certainly possible for Symbal's constituent models to capture similar biases as the original MLLM used for caption generation, we emphasize that the target tasks are different: the original MLLM is used for generation, whereas Symbal is used for verification. The verification procedure employed by Symbal, which involves evaluating alignment between image-text pairs and summarizing results, utilizes a fundamentally different reasoning process when compared to open-ended caption generation; as a result, this procedure is likely to be more robust to shared biases. This claim is supported by several recent works, which demonstrate that models prone to errors during free-form generation are still highly effective at verification (e.g. S Lee et al., NAACL 2024; J Guan et al. NAACL 2024); these studies suggest that even when identical model backbones are used for generation and verification, shared biases have a limited impact on verification abilities.
>
> For a more in-depth comparison of scorers (as well as other design choices), we refer the reviewer to Tables 4 and 5 in Appendix D.
>
> We again thank Reviewer LMQG for their review of our manuscript and their positive overall assessment of our work. We hope that the above responses adequately address all concerns.

---

### Official Review · Reviewer_6qV7 · 2025-10-30

**Soundness:** 4
**Presentation:** 3
**Contribution:** 4
**Rating:** 6
**Confidence:** 4

**Summary:**

This paper introduces "systematic misalignment detection," a new task for finding recurring errors in captions generated by Multimodal Large Language Models (MLLMs). A systematic misalignment is when a model repeatedly hallucinates a specific textual concept (e.g., "cardiomegaly") when a certain visual feature is present (e.g., "pacemaker").


I am a research conference reviewer. I will write a review for you, keeping it simple and human-like.

Summary This paper introduces "systematic misalignment detection," a new task for finding recurring errors in captions generated by Multimodal Large Language Models (MLLMs). A systematic misalignment is when a model repeatedly hallucinates a specific textual concept (e.g., "cardiomegaly") when a certain visual feature is present (e.g., "pacemaker").

To support this task, the authors create SymbalBENCH, a benchmark with 420 datasets (natural and medical images) containing known, injected systematic errors. They also propose Symbal, an automated two-stage method to detect these errors. Stage 1 groups and scores textual facts to find the likely error, and Stage 2 does the same for images to find the associated visual trigger.

**Strengths:**

1. Defining "systematic misalignment" as a distinct, detectable type of error is a great idea. It moves beyond just individual hallucinations to finding deeper model biases, which is very useful for the community.

2. The two-stage SYMBAL method (detect text error first, then visual trigger) is logical and effective. Breaking it down into grouping, scoring, and summarizing subtasks makes it modular and easy to improve.

3. Showing it can find actual systematic errors in Llava-1.5 (like hallucinations of "handbags" near "buses") validates the practical utility of the method.

**Weaknesses:**

1. Synthetic Benchmark: While necessary for a new task, the main evaluation is on synthetically injected errors. Real-world systematic errors might be subtler or more complex than the injected "object A causes hallucination of object B" pattern.

2. The method involves multiple steps of embedding, clustering, and calling large LLMs/MLLMs (like Qwen-72B) for every dataset. This might be slow or expensive for very large real-world datasets.

3. The statement that "SYMBALBENCH consists of a total of 420 vision-language datasets" is repeated frequently throughout the paper. This seems like an overstatement, as the methodology clarifies: "We repeat this procedure across a range of possible options for f and g, yielding 420 vision-language datasets with annotated systematic misalignments." (L232) In reality, only 2 base datasets (COCO and MIMIC-CXR) are used to create different versions with synthetic errors injected, which is far from the initial promise of 420 distinct datasets.

4. It is not clear to me, given a dataset, how many (f,g) pairs can be derived from it. A VLM can have several visual biases that result in wrong textual descriptions.

**Questions:**

1. How does the method perform if there are multiple different systematic misalignments in the same dataset? Can it detect more than one, or does it just find the strongest signal?

2. Have you tried SYMBAL on datasets where the visual trigger is more abstract (e.g., a specific lighting condition or camera angle causing a textual error) rather than just another object?

3. What is the typical runtime and cost to run the full SYMBAL pipeline on one of the benchmark datasets?

---

> ### Author Response · Authors · 2025-11-26
> **Response to Reviewer 6QV7 [1/2]**
>
> We thank Reviewer 6QV7 for reviewing our work and for providing helpful feedback.
>
> > **[Q1] Benchmark composition. “Synthetic Benchmark: While necessary for a new task, the main evaluation is on synthetically injected errors. Real-world systematic errors might be subtler or more complex than the injected "object A causes hallucination of object B" pattern.”**
>
> We refer the reviewer to General Response [Q1], where we address this point.
>
> > **[Q2] Computational cost. “The method involves multiple steps of embedding, clustering, and calling large LLMs/MLLMs (like Qwen-72B) for every dataset. This might be slow or expensive for very large real-world datasets…What is the typical runtime and cost to run the full SYMBAL pipeline on one of the benchmark datasets?”**
>
> As noted by the reviewer, Symbal does involve a multi-stage procedure with several calls to external models; however, our approach offers the key advantage of enabling fully-automated analysis of systematic misalignments, without requiring humans in the loop. Additionally, we emphasize that Symbal was designed to be computationally inexpensive. We use a maximum of 2 NVIDIA H100 80GB GPUs for all experiments, and we optimize our implementations of each subtask using the Faiss and vLLM packages.
>
> In response to your question, we evaluate Symbal's computational costs on a single natural image dataset in the SymbalBench benchmark. The total runtime for all subtasks is under 5 minutes. For Stage 1 of Symbal (detecting erroneous textual facts), grouping semantically-similar facts requires 20.8 seconds, alignment scoring requires 4.4 minutes, and summarization requires 1.5 seconds. For Stage 2 of Symbal (detecting associated visual features), grouping semantically-similar images requires 3.0 seconds, alignment scoring requires 0 seconds (as we use the cached alignment scores from Stage 1), and summarization requires 2.6 seconds. Our results demonstrate that our proposed approach operates inexpensively.
>
>
> > **[Q3] Number of datasets. “The statement that "SYMBALBENCH consists of a total of 420 vision-language datasets" is repeated frequently throughout the paper. This seems like an overstatement, as the methodology clarifies: "We repeat this procedure across a range of possible options for f and g, yielding 420 vision-language datasets with annotated systematic misalignments." (L232) In reality, only 2 base datasets (COCO and MIMIC-CXR) are used to create different versions with synthetic errors injected, which is far from the initial promise of 420 distinct datasets.”**
>
> SymbalBench does indeed include 420 vision-language datasets. Specifically, 360 of these datasets are derived from COCO, with each dataset consisting of 4349 image-text samples. The remaining 60 datasets are derived from MIMIC-CXR, with each dataset consisting of 2233 image-text samples.
>
> We view our use of just two base datasets (COCO and MIMIC-CXR) as a strength rather than a limitation, since this design choice allows us to solely vary the injected systematic misalignment while precisely controlling all other factors. This enables accurate comparisons across methods; for instance, as shown in Figure 4, we can explore how parameters like visual feature size and association strength affect performance, leaving all other dataset-related factors controlled.
>
> We also emphasize that while the base datasets remain similar, SymbalBench was explicitly designed to include diverse and challenging systematic misalignments. In particular, SymbalBench covers 2 domains, 156 ground-truth systematic misalignments ($f$,$g$) obtained from multiple sampling strategies, 3 association strengths, and visual features with varied sizes.  We refer the reviewer to Appendix Section B, where the composition of SymbalBench is discussed in detail.
>
> [Continued in next comment]

---

> > ### Author Response · Authors · 2025-11-26
> > **Response to Reviewer 6QV7 [2/2]**
> >
> > > **[Q4] Multiple systematic misalignments. “It is not clear to me, given a dataset, how many (f,g) pairs can be derived from it. A VLM can have several visual biases that result in wrong textual descriptions…How does the method perform if there are multiple different systematic misalignments in the same dataset? Can it detect more than one, or does it just find the strongest signal?”**
> >
> > Thank you for this question, and we agree that real-world datasets are likely to include multiple systematic misalignments. Symbal can be trivially extended to such settings. Specifically, Stage 1 of Symbal (Section 5.1) involves predicting the erroneous textual fact $\hat{f}$; here, rather than summarizing the single top ranked group of facts into a unifying concept, we can simply consider the top-k ranked groups instead. This will result in multiple predicted textual facts $\hat{f}_1$, $\hat{f}_2$,..., $\hat{f}_k$, each representing a distinct recurring textual error in the dataset. Stage 2 of Symbal can then be implemented as described in Section 5.2, taking into account each predicted textual fact; this will result in associated visual features $\hat{g}_1$, $\hat{g}_2$, ..., $\hat{g}_k$. Ultimately, at the conclusion of this procedure, Symbal will predict multiple systematic misalignments {($\hat{f}_i$, $\hat{g}_i$)} where $i$ ranges from 1 to $k$.
> >
> > In order to empirically demonstrate that Symbal is capable of identifying multiple systematic misalignments, we have updated our manuscript with substantial extensions to our real-world evaluations. In particular, we show that Symbal can accurately detect multiple real-world systematic misalignments in captions generated by Llava1.5-7B; in Figure 8, we demonstrate with both qualitative and quantitative analyses that these Symbal-predicted misalignments are accurate. We also show that Symbal can accurately detect multiple systematic misalignments in ShareGPT4V, an off-the-shelf dataset with MLLM-generated captions (Figure 10). We refer the reviewer to Section 6.4, Appendix Section E, Figure 8, and Figure 10 of our updated manuscript, where these extended evaluations are described in detail.
> >
> > > **[Q5] Abstract visual triggers. “Have you tried SYMBAL on datasets where the visual trigger is more abstract (e.g., a specific lighting condition or camera angle causing a textual error) rather than just another object?”**
> >
> > ​​Yes, Symbal is capable of identifying abstract systematic misalignments that extend beyond objects; in particular, the use of LLM-based and MLLM-based summarizers provides Symbal with the flexibility needed to identify diverse types of systematic misalignments.
> >
> > The real-world example provided as Example 5 in Appendix Section E is an example of one such misalignment. Here, in captions generated by LlavaOneVision-7B, Symbal detects that erroneous references to text ($\hat{f}$) in captions are often systematically associated with the presence of a sign ($\hat{g}$) in a scene. The visual examples in Figure 9 Row 2 validate this prediction, demonstrating that LlavaOneVision-7B repeatedly generates erroneous captions when signs are present; for instance, LlavaOneVision-7B interprets a handwritten sign stating "Have No Fear" as "Hate No Fear". Symbal is able to accurately identify this subtle misalignment, where $\hat{f}$ is not an object and $\hat{g}$ occurs in diverse forms.
> >
> > We do note that the majority of systematic misalignments identified by Symbal in real-world settings are object-related; this is in line with findings reported in prior works, which have found object-related errors to be particularly frequent in MLLM-generated captions (e.g. Li et al., EMNLP 2023).
> >
> > We again thank Reviewer 6QV7 for their review of our manuscript and their positive overall assessment of our work. We hope that the above responses adequately address all concerns.

---

### Official Review · Reviewer_VsQa · 2025-10-31

**Soundness:** 2
**Presentation:** 4
**Contribution:** 3
**Rating:** 4
**Confidence:** 4

**Summary:**

This paper introduces a new task called "systematic misalignment detection". This task aims to identify recurring errors in MLLM-generated captions that are strongly associated with the presence of a specific visual feature in the paired image. The authors present two key contributions: (1) SYMBALBENCH, the first benchmark for this task, which consists of 420 vision-language datasets with synthetically injected, annotated systematic misalignments across natural (COCO) and medical (MIMIC-CXR) domains. (2) SYMBAL, a novel, structured, dual-stage method to solve this task. The paper reports that SYMBAL correctly identifies misalignments in 63.8% of datasets (Acc@5, reference-free, open-ended), claiming this is a nearly 4x improvement over direct-prompting baselines.

**Strengths:**

1. The primary strength is the introduction of the "systematic misalignment detection" task. This is a new, well-motivated, and critical problem for the community as it addresses the need to audit MLLM-generated data at scale , particularly in high-stakes domains like medicine.
2. The paper contributes SYMBALBENCH, the first benchmark specifically designed for this task. This is a valuable resource for the community.
3. The paper is exceptionally clear and well-written. The problem is motivated compellingly, and the method is described in unambiguous detail.

**Weaknesses:**

1. The method's strong quantitative results (63.8% Acc@5) are derived entirely from the synthetic SYMBALBENCH, where errors were injected by the authors using a known procedure. This evaluation on a "clean" problem provides no guarantee of performance on the "messy" and subtle misalignments that may occur organically in the wild.
2. The "real-world" case study in Appendix E is insufficient. It consists of only three qualitative examples of errors found in LLaVA 1.5 . This is anecdotal evidence, not rigorous validation. There is no quantitative analysis (e.g., precision/recall on a human-annotated set of real errors) to support the claim of real-world utility.
3. The method appears fragile. It relies on K-Means with Silhouette distance to automatically select K, a notoriously unstable process. Furthermore, the "summarizing the top-ranked group" step  suggests a "winner-take-all" approach that can likely only detect one systematic error per dataset, which seems like a major limitation for real-world auditing.

**Questions:**

Please respond to the weaknesses I mentioned above.

---

> ### Author Response · Authors · 2025-11-26
> **Response to Reviewer VSQA**
>
> We thank Reviewer VSQA for reviewing our work and for providing helpful feedback.
>
> > **[Q1] Benchmark composition. “The method's strong quantitative results (63.8% Acc@5) are derived entirely from the synthetic SYMBALBENCH, where errors were injected by the authors using a known procedure. This evaluation on a "clean" problem provides no guarantee of performance on the "messy" and subtle misalignments that may occur organically in the wild.”**
>
> We refer the reviewer to General Response [Q1], where we address this point.
>
> > **[Q2] Real-world case studies. “The "real-world" case study in Appendix E is insufficient. It consists of only three qualitative examples of errors found in LLaVA 1.5 . This is anecdotal evidence, not rigorous validation. There is no quantitative analysis (e.g., precision/recall on a human-annotated set of real errors) to support the claim of real-world utility.”**
>
> We refer the reviewer to General Response [Q2], where we address this point.
>
> > **[Q3] Clarification on method. “It relies on K-Means with Silhouette distance to automatically select K, a notoriously unstable process. Furthermore, the "summarizing the top-ranked group" step suggests a "winner-take-all" approach that can likely only detect one systematic error per dataset, which seems like a major limitation for real-world auditing.”**
>
> **Clustering approach:** Symbal utilizes a clustering procedure in order to identify textual facts and visual features that occur consistently throughout the dataset. We select the value of K (number of clusters) using an automated approach, where we sweep across a wide range of potential cluster numbers and select the optimal value using Silhouette distance. This procedure offers the critical advantage of automating the selection of K, avoiding the need for users to manually set this parameter. We demonstrate empirically that this procedure works effectively across diverse dataset compositions and domains, both in SymbalBench and in the real world. We also emphasize that this procedure is in line with several prior works that have also utilized clustering objectives with automated selection of K when performing systematic error detection (e.g. Sohoni et al., NeurIPS 2022).
>
> **Settings with multiple systematic misalignments:** Thank you for raising this point, and we agree that real-world datasets are likely to include multiple systematic misalignments. Symbal can be trivially extended to such settings. Specifically, Stage 1 of Symbal (Section 5.1) involves predicting the erroneous textual fact $\hat{f}$; here, rather than summarizing the single top ranked group of facts into a unifying concept, we can simply consider the top-k ranked groups instead. This will result in multiple predicted textual facts $\hat{f}_1$, $\hat{f}_2$,..., $\hat{f}_k$, each representing a distinct recurring textual error in the dataset. Stage 2 of Symbal can then be implemented as described in Section 5.2, taking into account each predicted textual fact; this will result in associated visual features $\hat{g}_1$, $\hat{g}_2$, ..., $\hat{g}_k$. Ultimately, at the conclusion of this procedure, Symbal will predict multiple systematic misalignments {($\hat{f}_i$, $\hat{g}_i$)} where $i$ ranges from 1 to $k$.
>
> In order to empirically demonstrate that Symbal is capable of identifying multiple systematic misalignments, we have updated our manuscript with substantial extensions to our real-world evaluations. In particular, we show that Symbal can accurately detect multiple real-world systematic misalignments in captions generated by Llava1.5-7B; in Figure 8, we demonstrate with both qualitative and quantitative analyses that these Symbal-predicted misalignments are accurate. We also show that Symbal can accurately detect multiple systematic misalignments in ShareGPT4V, an off-the-shelf dataset with MLLM-generated captions (Figure 10). We refer the reviewer to Section 6.4, Appendix Section E, Figure 8, and Figure 10 of our updated manuscript, where these extended evaluations are described in detail.
>
> We again thank Reviewer VSQA for their review of our manuscript. We hope that the above responses adequately address all concerns.

---

### Official Review · Reviewer_QdbW · 2025-11-03

**Soundness:** 2
**Presentation:** 1
**Contribution:** 2
**Rating:** 2
**Confidence:** 4

**Summary:**

This paper focuses on detecting systematic misalignments (that is hallucinations or inaccuracies) in generated captions. First, the paper proposes the synthetic benchmark, “SymbalBench”, to analyze the different methods to detect such hallucinations. Authors found that existing MLLMs fail on hallucination detection. And they propose Symbal as a hallucination detection method for this task which outperforms the selected baselines.

**Strengths:**

* The paper focuses on the hallucination (systematic misalignments) for the vision-laguage models, which is a very important problem statement.
* SymbalBench shows that existing LLMs/VLMs are poor at detecting such hallucinations (likely because they are the ones creating hallucinations).
* Symbal is an empirical low-compute (potentially) method that outperforms the baselines.
* Additionally, benchmark focuses  on the real and medical domains.

**Weaknesses:**

* The paper is poorly written and hard to follow. For instance, it is unclear if there are 420 datasets or 420 examples in the SymbalBench!
* Additionally, Symbal benchmark seems illposed. Specifically, the goal is to evaluate the MLLM generated captions. While the proposed benchmark creates the synthetic task pairs.
* Missing human evaluations. It is clear that Symbal outperforms the zero-shot baselines. However, it is not clear whether it will generalize to real-life scenarios. I would advise to conduct the human evaluations and compare with Symbal (and baselines) on generated captions from recent advance VLMs.

**Questions:**

See the weaknesses.

---

> ### Author Response · Authors · 2025-11-26
> **Response to Reviewer QDBW**
>
> We thank Reviewer QDBW for reviewing our work and for providing helpful feedback.
>
> >**[Q1] Clarification on SymbalBench. “It is unclear if there are 420 datasets or 420 examples in the SymbalBench”**
>
> SymbalBench includes a total of **420 datasets**. Specifically, 360 of these datasets are derived from COCO, with each dataset consisting of 4349 image-text samples. The remaining 60 datasets are derived from MIMIC-CXR, with each dataset consisting of 2233 image-text samples.
>
> Each dataset is paired with a ground-truth annotation ($f$, $g$) indicating the systematic misalignment, where $f$ represents the erroneous textual fact and $g$ represents the associated visual feature. Our approach Symbal accepts an entire dataset (with thousands of image-text samples) as input and is tasked with predicting the systematic misalignment ($\hat{f}$, $\hat{g}$) as output.
>
> > **[Q2] Benchmark composition. “Specifically, the goal is to evaluate the MLLM generated captions. While the proposed benchmark creates the synthetic task pairs.”**
>
> We refer the reviewer to General Response [Q1], where we address this point.
>
> > **[Q3] Additional evaluations. “However, it is not clear whether [Symbal] will generalize to real-life scenarios. I would advise to conduct the human evaluations and compare with Symbal (and baselines) on generated captions from recent advance VLMs.”**
>
> We refer the reviewer to General Response [Q2], where we address this point.
>
> We again thank Reviewer QDBW for their review of our manuscript. We hope that the above responses adequately address all concerns.

---

### Author Response · Authors · 2025-11-26
**General Response [1/5]**

We thank the reviewers for their thoughtful review of our manuscript. We were encouraged to see that reviewers found our novel task to be "a great idea" that is "important", "well-motivated", and "[of] significant value" (Reviewers QDBW, VSQA, 6QV7, LMQG); our method to be "logical", "intuitive", and "low-compute" (Reviewers QDBW, 6QV7, LMQG); and our benchmark to be a valuable resource for the community (Reviewers VSQA, 6QV7). Reviewers also noted that our paper was "exceptionally clear and well-written" (Reviewer VSQA).

In response to feedback, we provide a general response here to points raised by multiple reviewers, individual responses below to address each reviewer's concerns, and an updated manuscript.

> **[Q1] Reviewers QDBW, VSQA, 6QV7, LMQG asked for additional discussion with respect to the synthetic nature of SymbalBench.**

We thank the reviewers for raising this point. Below, we (1) motivate the need for SymbalBench, (2) clarify our benchmark creation procedure, which is explicitly designed to represent the types of subtle, complex misalignments likely to emerge in the wild, and (3) perform additional evaluations that extend beyond SymbalBench to real-world settings.

**Motivation behind SymbalBench:** The key challenge behind evaluating methods like Symbal on real-world vision-language datasets is that ground-truth systematic misalignments are unknown. Previous works have explored misalignments at the per-sample level by comparing model-generated captions with human-written captions; however, we emphasize that our work focuses on **systematic** misalignments, a novel task for which there are no existing human-annotated datasets. Moreover, collecting human annotations for a task at this scale, where datasets include thousands of images paired with information-dense captions, is simply intractable.

Thus, without access to ground-truth annotations, it becomes difficult (1) to determine whether misalignments identified by a method like Symbal are accurate and (2) to quantitatively compare results across multiple methods. SymbalBench is designed to address this challenge. Specifically, SymbalBench utilizes an automated method to inject pre-defined systematic misalignments into base vision-language datasets, yielding a test bed where ground-truth annotations are available. The automated nature of our approach provides several key advantages, including (1) the ability to operate at scale by generating hundreds of datasets, (2) the presence of ground-truth labels that are guaranteed to be accurate, and (3) the ability to extend to specialized domains like medical imaging.

We also emphasize that SymbalBench extends an established line of research; many recent benchmarks have used automated methods to generate datasets with specific pre-defined characteristics, such as injected errors. Some popular examples include the FOIL dataset (Shekhar et al., ACL 2017), the VisDiff dataset (Dunlap et al., CVPR 2024), and the Domino evaluation framework (Eyuboglu et al., ICLR 2022).

[Continued in next comment]

---

> ### Author Response · Authors · 2025-11-26
> **General Response [2/5]**
>
> **Benchmark creation procedure:** Our automated procedure for injecting pre-defined systematic misalignments is explicitly designed to represent the types of subtle, complex misalignments that are likely to emerge in the wild. Specifically, our benchmark covers:
> - **Multiple domains:** SymbalBench encompasses both natural image and medical image data, representing two distinct domains. Medical imaging data represents a complex and specialized setting, where detection of systematic misalignments is particularly critical prior to real-world deployment.
> - **Diverse misalignments that are plausible in real-world settings:**
>   - For our natural image datasets, we use three possible sampling strategies when predefining $f$ and $g$, which is designed to capture a range of possible error patterns likely to emerge in real-world MLLM-generated captions. Specifically, we first sample $g$ from the set of 80 object categories present in the dataset. Then, we sample $f$ from the set of 80 object categories (such that $f \neq g$) utilizing the following three sampling strategies: (1) random, where $f$ is sampled randomly, (2) popular, where $f$ is sampled from the list of the top-ten most popular objects in the COCO training set, and (3) adversarial, where $f$ is the object that most commonly co-occurs with $g$ in the COCO training set. For instance, this procedure yields a dataset in SymbalBench with ground-truth systematic misalignment $f$=surfboard and $g$=airplane (random), a dataset with ground-truth systematic misalignment $f$=bottle and $g$=airplane (popular), and a dataset with ground-truth systematic misalignment $f$=person and $g$=airplane (adversarial). These sampling strategies are motivated by prior work (Li et al., EMNLP 2023).
>   - For our medical image datasets, we sample $f$ from a set of five disease categories selected from the commonly-used CheXpert annotation list (Irvin et al., 2019): cardiomegaly, pneumothorax, atelectasis, pleural effusion, and edema. We sample $g$ from a set of five medical devices: pacemaker, chest tube, endotracheal tube, surgical clips, sternotomy wires. We select these options for $f$ and $g$ since medical devices often co-occur with diseases, yet there is no deterministic, universal link. Prior works (e.g. Oakden-Rayner et al., CHIL 2020) have demonstrated that models often learn spurious associations between devices and diseases, meaning that such errors are highly plausible in MLLM-generated reports.
> - **Varying strengths of the injected systematic misalignment:** In real-world settings, systematic misalignments exhibit variations in strength. In order to reflect this, we consider three possible levels of association between the erroneous textual fact $f$ and visual feature $g$ in our datasets: low association (Cramer’s V = 0.3), moderate association (Cramer’s V = 0.6), and high association (Cramer’s V = 0.9).
> - **Visual features of varying sizes:** Visual features in real-world datasets exhibit varying sizes, and in particular, small features present a particularly challenging setting. SymbalBench exhibits substantial coverage across varying sizes of visual feature $g$, as shown in Figure 6.
>
> We provide a detailed breakdown of the composition of SymbalBench in Figures 5 and 6. Stratified performance metrics across each of these categories are provided in Figure 4 (misalignment strength and feature sizes) and Figure 7 (sampling strategy). Table 3 provides a comprehensive list of all ground-truth systematic misalignment annotations (i.e. ($f$, $g$)) included in SymbalBench.
>
> In summary, by including multiple domains, diverse misalignments, varying strengths of the injected systematic misalignment, and visual features of varying sizes, we are able to construct a comprehensive benchmark that reflects the types of misalignments likely to emerge in real-world settings.
>
> **Extending to real-world misalignments:** In order to demonstrate the utility of our proposed method beyond SymbalBench, we have updated our paper with substantial extensions to our real-world evaluations. We show that Symbal is able to surface complex systematic misalignments in real-world settings; for example, in captions generated by LlavaOneVision-7B, Symbal detects that erroneous descriptions of text ($\hat{f}$) in captions are often systematically associated with the presence of a "sign” ($\hat{g}$) in a scene (Figure 9, Row 2). This systematic misalignment points to limitations with the OCR capabilities of LlavaOneVision-7B, suggesting that the presence of text-based signage in an image is likely to result in errors in the generated caption. Our extended evaluations can be found in Section 6.4, Appendix Section E, Figure 8, Figure 9, and Figure 10 of our updated manuscript; we describe these extensions in further detail in our response to [Q2] below.
>
> [Continued in next comment]

---

> > ### Author Response · Authors · 2025-11-26
> > **General Response [3/5]**
> >
> > > **[Q2] Reviewers QDBW and VSQA asked for additional evaluations of Symbal in real-world settings.**
> >
> > Thank you for this suggestion. In response to your point, we have substantially extended our real-world evaluations, which can be found in Section 6.4, Appendix Section E, Figure 8, Figure 9, and Figure 10 of our updated manuscript.
> >
> > We emphasize that since ground-truth systematic misalignments in real-world settings are unknown, assessing the accuracy of results poses a computational challenge. Collecting human annotations is simply intractable given the large size of image-text datasets, the information density of MLLM-generated captions, and the potential for multiple systematic misalignments in each dataset. Our benchmark SymbalBench, discussed in Section 3, was specifically designed to address these challenges, enabling large-scale quantitative evaluations of methods designed for the systematic misalignment detection task. However, in response to your suggestion, we have extended our evaluations to real-world settings beyond SymbalBench using the following procedure. First, we qualitatively validate the existence of Symbal-identified systematic misalignments with visual analysis. Second, we quantitatively validate whether a link between erroneous fact $\hat{f}$ and visual feature $\hat{g}$ truly exists; to this end, we measure whether model-generated captions are indeed more likely to include erroneous references to $\hat{f}$ when $\hat{g}$ is present compared to when $\hat{g}$ is absent. In order to perform this evaluation, we use a state-of-the-art open-set object detector (Minderer et al., 2024) to annotate the presence of $\hat{g}$ in each image, and we use our top-performing alignment scorer (vision-language scorer with Qwen-72B) to annotate erroneous references to $\hat{f}$ in each caption. Additional analysis is provided below confirming that this automated annotation procedure aligns closely with human judgments.
> >
> > We perform two lines of evaluation, where we (1) use Symbal to evaluate captions generated by off-the-shelf MLLMs and (2) use Symbal to audit ShareGPT4V, an open-source vision-language dataset with model-generated captions.
> >
> > **Real-World Evaluation 1: Quantitative and qualitative analyses across four MLLMs demonstrate that Symbal can accurately surface systematic misalignments in model-generated captions.** We use Symbal to analyze captions generated by four real-world off-the-shelf MLLMs: Llava1.5-7B, Llava1.5-13B, AyaVision-8B, and LlavaOneVision-7B. We utilize each model to generate captions for the COCO dataset (2017 val split); we then apply Symbal (reference-free, open-ended) to predict systematic misalignments ($\hat{f}$, $\hat{g}$). Below, we list several Symbal-identified systematic misalignments along with associated validation.
> > - **Example 1:** In captions generated by Llava1.5-7B, Symbal detects that erroneous references to a “TV” ($\hat{f}$) in captions are often systematically associated with the presence of a “desk”, “computer monitor”, and/or “keyboard” ($\hat{g}$) in the scene. We provide visual examples of image-caption pairs with the Symbal-identified systematic misalignment in Figure 8 (Row 1). Quantitatively, our analysis finds that erroneous references to a “TV” in model-generated captions are indeed 13.5 times more likely when a “desk” is present in the image compared to when a “desk” is absent, validating the Symbal prediction.
> > - **Example 2:** In captions generated by Llava1.5-7B, Symbal detects that erroneous references to a "handbag” or a "handbag on the ground” ($\hat{f}$) in captions are often systematically associated with the presence of a "bus” ($\hat{g}$) in a scene. We provide visual examples of image-caption pairs with the Symbal-identified systematic misalignment in Figure 8 (Row 2). Quantitatively, our analysis finds that erroneous references to a "handbag” in model-generated captions are indeed 3.1 times more likely when a "bus” is present in the image compared to when a "bus” is absent, validating the Symbal prediction.
> > - **Example 3:** In captions generated by Llava1.5-7B, Symbal detects that erroneous references to a "chair” ($\hat{f}$) in captions are often systematically associated with the presence of a "television” ($\hat{g}$) in a scene. We provide visual examples of image-caption pairs with the Symbal-identified systematic misalignment in Figure 8 (Row 3). Quantitatively, our analysis finds that erroneous references to a "chair” in model-generated captions are indeed 3.1 times more likely when a "television” is present in the image compared to when a "television” is absent, validating the Symbal prediction.
> >
> > [Continued in next comment]

---

> ### Author Response · Authors · 2025-11-26
> **General Response [4/5]**
>
> - **Example 4:** In captions generated by Llava1.5-13B, Symbal detects that erroneous references to a "TV” ($\hat{f}$) in captions are often systematically associated with the presence of a "computer monitor”, "keyboard”, and/or "mouse” ($\hat{g}$) in a scene. **Interestingly, this systematic misalignment is nearly identical to one that exists in Llava1.5-7B-generated captions (see Example 1), suggesting that solely increasing the scale of the underlying MLLM is insufficient for resolving systematic misalignments.** We provide visual examples of image-caption pairs with the Symbal-identified systematic misalignment in Figure 9 (Row 1). Quantitatively, our analysis finds that erroneous references to a "TV” in model-generated captions are indeed 22.2 times more likely when a "computer monitor” is present in the image compared to when a "computer monitor” is absent, validating the Symbal prediction.
> - **Example 5:** In captions generated by LlavaOneVision-7B, Symbal detects that erroneous references to "text” ($\hat{f}$) in captions are often systematically associated with the presence of a "sign” ($\hat{g}$) in a scene. **This systematic misalignment suggests that LlavaOneVision-7B struggles with OCR capabilities, where the presence of text-based signage in an image is likely to result in errors in the generated caption.** We provide visual examples of image-caption pairs with the Symbal-identified systematic misalignment in Figure 9 (Row 2). Quantitatively, our analysis finds that erroneous references to "text” in model-generated captions are indeed 4.6 times more likely when a "sign” is present in the image compared to when a "sign” is absent, validating the Symbal prediction.
> - **Example 6:** In captions generated by AyaVision-8B, Symbal detects that erroneous references to a "vase” ($\hat{f}$) in captions are often systematically associated with the presence of a "couch” ($\hat{g}$) in a scene. We provide visual examples of image-caption pairs with the Symbal-identified systematic misalignment in Figure 9 (Row 3). Quantitatively, our analysis finds that erroneous references to a "vase” in model-generated captions are indeed 17.7 times more likely when a "couch” is present in the image compared to when a "couch” is absent, validating the Symbal prediction.
>
> Overall, across all examples of Symbal-identified systematic misalignments provided above, we find that erroneous references to $\hat{f}$ are substantially more likely when $\hat{g}$ is present in the image compared to when $\hat{g}$ is absent. This analysis validates discovered misalignments by demonstrating that links between Symbal-identified erroneous textual fact $\hat{f}$ and Symbal-identified visual feature $\hat{g}$ do indeed exist.
>
> As stated previously, our quantitative validation procedure relies on automated annotation methods in order to enable evaluation at scale; in particular, we leverage Qwen-72B in order to annotate erroneous references to $\hat{f}$ in each caption. We find that these generated annotations align closely with human judgments. Given the set of 215 images in the dataset containing a “bus", we tasked a human reader with identifying whether each Llava1.5-7B-generated caption contained an erroneous reference to a “handbag" and/or “handbag on the ground" (Example 2). Human judgments aligned perfectly with Qwen-72B predictions in 96.3% of cases (Cohen's kappa = 0.86).
>
> [Continued in next comment]

---

> > ### Author Response · Authors · 2025-11-26
> > **General Response [5/5]**
> >
> > **Real-World Evaluation 2: Symbal is a powerful tool for auditing open-source vision-language datasets.** We use Symbal to analyze ShareGPT4V, an open-source image dataset with MLLM-generated captions commonly used as a pretraining dataset for vision-language models. We sample a subset of 10k image-caption pairs from the ShareGPT4V dataset, and we then apply Symbal (reference-free, open-ended) to predict systematic misalignments ($\hat{f}$, $\hat{g}$). Symbal identifies several systematic misalignments:
> > - **Example 7:** Symbal detects that erroneous references to a "white tablecloth” ($\hat{f}$) in captions are often systematically associated with the presence of a "table”, "cake”, and/or "people” ($\hat{g}$) in the scene. We provide visual examples of image-caption pairs with the Symbal-identified systematic misalignment in Figure 10 (Row 1). Quantitatively, our analysis finds that erroneous references to a "white tablecloth” in model-generated captions are indeed 17.2 times more likely when a "table” is present in the image compared to when a "table” is absent, validating the Symbal prediction.
> > - **Example 8:** Symbal detects that erroneous references to a "printer” ($\hat{f}$) in captions are often systematically associated with the presence of a "computer monitor” ($\hat{g}$) in a scene. We provide visual examples of image-caption pairs with the Symbal-identified systematic misalignment in Figure 10 (Row 2). Quantitatively, our analysis finds that erroneous references to a "printer” in model-generated captions are indeed 121 times more likely when a "computer monitor” is present in the image compared to when a "computer monitor” is absent, validating the Symbal prediction.
> > - **Example 9:** Symbal detects that erroneous references to a "black phone” ($\hat{f}$) in captions are often systematically associated with the presence of a "laptop” ($\hat{g}$) in a scene. We provide visual examples of image-caption pairs with the Symbal-identified systematic misalignment in Figure 10 (Row 3). Quantitatively, our analysis finds that erroneous references to a "black phone} in model-generated captions are indeed 48.5 times more likely when a "laptop” is present in the image compared to when a "laptop” is absent, validating the Symbal prediction.
> >
> > As large-scale datasets like ShareGPT4V become increasingly prevalent, it becomes critical for users to be aware of potential systematic misalignments, as these errors can propagate to trained models. Specifically, if a dataset contains a systematic misalignment between erroneous textual fact $\hat{f}$ and visual feature $\hat{g}$, models trained on the dataset are likely to learn spurious correlations between $\hat{f}$ and $\hat{g}$, leading to prediction errors at test-time.
> >
> > Ultimately, knowledge of systematic misalignments can aid users with understanding limitations of datasets with MLLM-generated captions as well as aid model developers with improving performance of MLLMs.
> >
> > We would again like to thank all reviewers for their time and feedback, and we hope that our responses adequately address all concerns.

---

### Meta-Review · Area_Chair_m87T · 2026-01-06

**Summary:**

This paper introduces the task of systematic misalignment detection for model-generated captions, along with a synthetic benchmark (SymbalBench) and a two-stage method (Symbal) designed to identify recurring captioning errors tied to specific visual features. Reviewers generally agreed that the problem is interesting, the paper is clearly written, and the proposed pipeline is logical and well engineered.

As it is being said, a decision to recommend rejection from AC given concerns shared across multiple reviews about both the problem formulation and the practical significance of the contribution.

**Reviewer Concerns:**

The strongest concern is that the task is defined and validated almost entirely through a synthetic benchmark where errors are injected using a known procedure. While the authors argue this is necessary for a new task, reviewers consistently questioned whether strong performance on this clean, controlled setting meaningfully reflects the messy, subtle, and heterogeneous misalignments that occur in real-world data. The added real-world evaluations, although helpful, remain largely qualitative and limited in scale, and do not fully address this gap and up to the NeurIPS quality bar.

AC agree with Rs that the main novelty lies in task naming and benchmark construction, while the method itself is a composition of existing components such as clustering, alignment scoring, and LLM-based summarization. Several reviewers noted that the gains appear to come from careful system design rather than from a new conceptual insight, and it remains unclear how much the approach advances our understanding of model misalignment beyond what existing auditing or bias-discovery tools already provide. The potential impact is thus fairly limited.

Finally, reviewers raised questions about the problem setting itself. It is not clear that reducing systematic misalignment detection to identifying a small number of dominant pairs per dataset captures the complexity of real-world captioning errors, where multiple overlapping biases, weak correlations, or abstract visual triggers may coexist. While the authors describe extensions to handle multiple misalignments, these remain under-explored, and the reliance on powerful off-the-shelf models for both detection and verification further blurs what is genuinely being discovered versus what is inferred by external models. This limits the excitement further.

Based on these concerns, I recommend rejection.

**Reviewer Scores:**

relatively inexperienced reviewer LmQg may lower its score given other reviewers' comments and aligning him/her self with the cohort.

---

### Decision · Program_Chairs · 2026-01-26

Reject